# OAM-TCD: A globally diverse dataset of high-resolution tree cover maps

**Josh Veitch-Michaelis**[1,2,6*]   **Andrew Cottam**[2]   **Daniella Schweizer**[2,3]   **Eben N. Broadbent**[4]
**David Dao**[2,5]   **Ce Zhang**[1,6]   **Angelica Almeyda Zambrano**[4]   **Simeon Max**[2,5]

[1]ETH Zurich   [2]Restor   [3]WSL   [4]University of Florida   [5]Gainforest   [6]University of Chicago
{josh,andrew}@restor.eco   daniella.schweizer@wsl.ch
{eben,aalmeyda}@ufl.edu   {david,simeon}@gainforest.net   cez@uchicago.edu

## Abstract

Accurately quantifying tree cover is an important metric for ecosystem monitoring and for assessing progress in restored sites. Recent works have shown that deep learning-based segmentation algorithms are capable of accurately mapping trees at country and continental scales using high-resolution aerial and satellite imagery. Mapping at high (ideally sub-meter) resolution is necessary to identify individual trees, however there are few open-access datasets containing instance level annotations and those that exist are small or not geographically diverse. We present a novel open-access dataset for individual tree crown delineation (TCD) in high-resolution aerial imagery sourced from OpenAerialMap (OAM). Our dataset, OAM-TCD, comprises 5072 2048x2048 px images at 10 cm/px resolution with associated human-labeled instance masks for over 280k individual and 56k groups of trees. By sampling imagery from around the world, we are able to better capture the diversity and morphology of trees in different terrestrial biomes and in both urban and natural environments. Using our dataset, we train reference instance and semantic segmentation models that compare favorably to existing state-of-the-art models. We assess performance through k-fold cross-validation and comparison with existing datasets; additionally we demonstrate compelling results on independent aerial imagery captured over Switzerland and compare to municipal tree inventories and LIDAR-derived canopy maps in the city of Zurich. Our dataset, models and training/benchmark code are publicly released under permissive open-source licenses: Creative Commons (majority CC BY 4.0), and Apache 2.0 respectively.

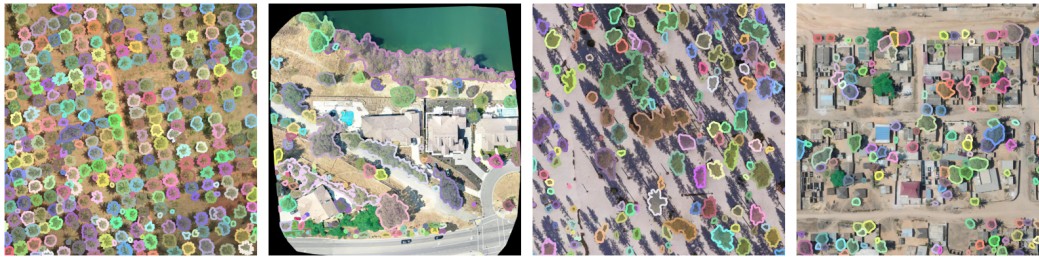

Figure 1: Annotation examples from the OAM-TCD dataset. Source images are 2048x2048 px tiles at 10 cm/px resolution. Individual instances are labelled with different colors. Annotators were instructed to label individual trees if possible, and otherwise label regions as groups. Image credit: contributors to Open Imagery Network, CC BY 4.0.

38th Conference on Neural Information Processing Systems (NeurIPS 2024) Track on Datasets and Benchmarks.

# 1 Introduction

It is estimated that 2.8 billion hectares of land on Earth is populated by tree cover [36]. In the context of climate change mitigation, while not a panacaea, forest restoration has been identified as one of the most effective methods for large scale carbon capture [12, 49, 56]. Trees are also important for biodiversity and, aside from the ecological benefits of responsible afforestation and reforestation, tree cover in urban environments has been shown to positively impact human health and well-being [28, 65, 81]. Mapping tree cover is relevant for a wide variety of domains and there is a corresponding need for transparent and globally effective monitoring [53].

Monitoring, in ecological contexts, is an umbrella term that encompasses a variety of remote sensing techniques that aim to characterise an ecosystem [52] in support of a goal, such as improving canopy coverage or reintroduction or preservation of a species. Quantifying tree canopy cover is typically only one component of a monitoring protocol and any changes in cover should be compared to baseline or reference conditions (such as other sites in the same ecoregion or biome type [61]).

Satellite imaging at the 1-10 m scale is routinely used for global tree mapping, but there are few open-access models and datasets that facilitate higher resolution mapping at the sub-meter scale. This is particularly relevant for assessing sparsely distributed trees outside forested areas which are not visible in low resolution images. Although individual trees may be visible at 1 m/px resolution, this is still challenging for instance segmentation where targets may only be a few pixels across. It is essential that any dataset used for global mapping is trained on equally diverse training data, which is a deficiency of most existing open-access tree segmentation datasets.

To address this challenge, we release a high-resolution, globally diverse, segmentation dataset for tree cover mapping. Our dataset, OAM-TCD, is derived from OpenAerialMap [8] and contains over 20k hectares of labelled image tiles at 10 cm/px resolution. Annotations include instance-level masks for over 280k trees and over 56k regions of closed canopy (tree groups). Alongside the dataset, we provide baseline segmentation models and an open-source training, prediction and reporting pipeline for processing arbitrarily large orthomosaic images.

## 1.1 Challenges in global tree detection

There are at least 60k confirmed species of tree in the world [22, 73], but there is no universally agreed-upon definition for which species are referred to as trees. Trees exhibit tremendous diversity in morphology; individuals from some species can easily be distinguished from the air using visible/RGB images, such as palms, but others cannot. For example in [80] and [11], to label trees in images of dense forest, annotators were provided with additional metadata like Canopy Height Models (CHMs) derived from Airborne Laser Scanning (ALS/LIDAR), hyperspectral imagery and contrast-adjusted RGB imagery.

We use the generic phrase tree *mapping* to refer to a myriad applications ranging from density estimation/counting and species distribution to canopy coverage and trait estimation. In our work, we are most concerned with identifying presence/absence of tree cover, a foundational data product which is required for other downstream analyses. In addition to natural forests, monitoring tree cover is also important in urban environments [13, 59]. Many cities record detailed inventories of municipally managed trees and, in response to climate change, have set targets for increased "green" cover; the city of Zurich for example has committed to improving crown area coverage from 17 % (2018) to 25%, measured via LIDAR [1].

To avoid ambiguity, a gold-standard dataset would contain species-level annotations for every individual plant, regardless of age or size. In practice, different monitoring campaigns may only consider vegetation that meets certain morphological requirements, depending on project goals; this is then reflected in what objects/species are labelled, and how. Not only does this make constructing a generic tree detection dataset difficult, but it also poses a problem for comparative purposes because of the differences in class definitions between existing datasets.

In datasets where species annotations are not provided, a common criterion for inclusion is vegetation height. Hansen et al [36] consider everything above 5 meters, Global Forest Watch modify this to also include 3-5 m tall vegetation with a crown diameter of at least 5 m and DeepForest [79] uses a minimum height of 3 m (and comparison to a LIDAR CHM).

## 1.2 Data modality and resolution

Tree coverage maps, such as those used to track deforestation, are largely facilitated by satellite imagery which is updated at regular intervals. Open access platforms like the European Space Agency (ESA) Copernicus/Sentinel [27, 64] missions provide frequent coverage of the globe at 10-30 m resolution (in most areas, 5 day re-visit); this analogous to NASA's Landsat program [83]. Both platforms are routinely used for global tree mapping and land cover analyses [2, 15, 17, 36, 47].

Several recent publications have demonstrated that country- and even continental-scale mapping is feasible at the meter and sub-meter scale [18, 50, 58, 66, 74]. These approaches are able to segment individual trees, and results suggest that sparse tree cover is likely underestimated in low resolution images [29]. However, the use of commercial data sources like Planet or Maxar imagery make these studies expensive and/or difficult to replicate, especially over such large areas. A notable exception is Norway's International Climate and Forest Initiative (NICFI) [7, 57] which provides funding for non-commercial Planet (< 5 m/px) image access over the tropics.

Imagery below 0.3 m/px is typically captured via aerial or Unmanned Aerial Vehicle (UAV, drone) surveys [21]. UAV data are readily available and offer extremely high resolution, < 0.1 m/px. Since normally only geo-referenced RGB information is available, we focus on detection models that do not require LIDAR. This inherently limits the analysis that can be performed by our models, without absolute CHMs or other spectral bands. Digital Surface Models (DSMs) are a by-product of photogrammetric processing (orthomosaic generation) and have been explored as a low-cost alternative to LIDAR for forest structure estimation [19, 33, 63], however DSMs were unavailable for our training data as OAM currently does not provide them.

## 2 Related work

Depending on monitoring requirements, tree detection in optical imagery can be considered a segmentation (semantic, instance, panoptic); object detection (bounding box); or keypoint detection task. In the pre- deep-learning era, until around 2012 [45], most methods for tree mapping employed classical computer vision approaches like local maxima extraction, region growing algorithms and template matching [42, 86]. The main shortcoming of these approaches is distribution shift, where models fail to generalise to inputs outside the training data domain [14, 44]. This is still an issue for contemporary approaches and there is evidence that training on multiple domains (sites, biomes) provides improved results compared to site-specific models trained only on local data [80]. We acknowledge that a large body of literature focuses on LIDAR and low-resolution satellite imaging, but given that our model does not use this data we focus on work that uses high-resolution (sub-meter) RGB imagery.

**Modelling**    Following the trend of other computer vision tasks, recent state-of-the-art methods for tree detection in aerial imagery rely on deep neural networks [48]. Classical methods are still used (and are effective)  [16, 54, 55, 68], but they suffer from the generalisation shortcomings described above. Detecting trees in homogeneous settings like plantations, or trees with distinct morphology (e.g. palms) is now relatively simple using off-the-shelf object detection pipelines, where models can benefit from few-shot learning [77] on a small quantity of representative examples [26, 34]; it is not uncommon to see reported accuracies of over 90% on test datasets in these scenarios.

Deep learning-based methods have tracked trends in model architecture popularity [24, 35, 51]. Most publications report using standard architectures: for example UNet for semantic segmentation [31, 50, 67, 71, 72] and RCNN-based models for object and instance segmentation [32, 37, 60, 69, 79, 84]. More recently, there has been increased interest in transformer-based models  [10, 34, 75] and general-purpose segmentation models like Segment Anything (SAM) [43, 62]. In the last year, several foundational models, trained on large quantities of geospatial data, have been released [4, 41, 74] and show potential for tree detection purposes.

**Datasets**    Despite continual and frequent advancements in model architectures, the open data landscape for high resolution tree detection is more limited. From the literature listed in the previous section, very few works have released their training data. One effort to collate open datasets from the literature is the MillionTrees project which aims to create a unified benchmark for tree detection across task types [5]. Still, the majority of the datasets listed on MillionTrees are limited to, at best, a

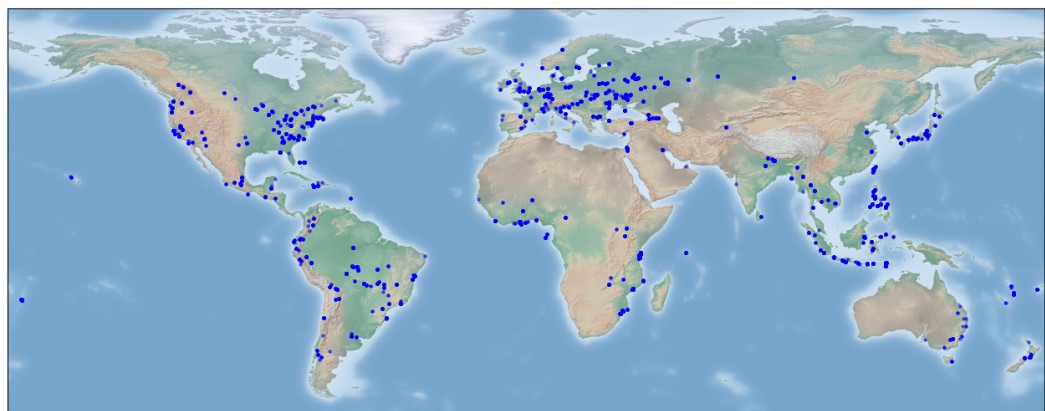

Figure 2: Geospatial distribution of imagery in the dataset. It is clear that some locations are under-represented, but among open-access data, we believe OAM-TCD is the most geographically diverse of its type. The lack of imagery from some regions is due to inherent biases in the data that are uploaded to OAM.

single country. Interestingly, almost all the datasets currently listed on MillionTrees were published in the last 2-3 years which suggests that attitudes to data release are improving within the community.

A notable open dataset is NeonTreeEvaluation [78], published alongside DeepForest, which is derived from the National Ecological Observation Network (NEON) [6] in the USA. NeonTreeEvaluation includes bounding-box training data from several distinct sites and includes aerial RGB, LIDAR and hyperspectral imaging. This dataset was used to train DeepForest and was released as a benchmark. Other relevant works like DetecTree2 (instance segmentation) [11] and AutoArborist [13] provide partial training data or require an access request.

As more governmental organisations release open-access geospatial data there is an opportunity to construct benchmark datasets without needing to acquire labels (for example LIDAR surveys, tree inventory, land-cover maps). In this work we combine a variety of data products that are published by the city of Zurich and Switzerland's federal mapping institute to assess our models. This is an approach used by others: DetecTree's [16] semantic segmentation model was trained on an earlier release of this imagery and [50] take advantage of state-acquired imagery over Finland and Denmark. Alongside NEON, other works have used NAIP (National Agricultural Imaging Program) data at 60 cm, also over the US [70]. Tree location databases may also be used as keypoint ground truth for density estimation or counting, but are less useful for area/coverage prediction where crown size is required.

To summarise, we find that (1) a large variety of ML models have been employed for tree detection; (2) most works do not share their data and most only evaluate on a small geographic region. We designed our dataset to address these issues, in particular the absence of a global instance-level tree detection dataset.

## 3 OAM-TCD Dataset

Our dataset - OAM-TCD - is derived from imagery obtained from OpenAerialMap (OAM) [8] and contains instance-level (polygon) annotations for individual and groups of trees. OAM is a repository of permissively licensed global aerial imagery, much of which is user-contributed from UAV-based surveys, alongside some public-domain satellite data (typically from post-disaster events). Sample image tiles from OAM-TCD are shown in Figure 1 and a map of the geospatial distribution of the entire dataset is shown in Figure 2. Our aim is to provide the community with an open dataset to support tree mapping in global contexts while allowing users to experiment with different mapping tasks. By providing polygon annotations, we provide more flexibility over datasets that only contain keypoints (crown center locations) or bounding boxes.

## 3.1 Stratification and diversity

Our dataset was constructed by repeatedly sampling OAM images from a 1-degree binned world map. Each image was split into tiles which were randomly sampled with some simple filtering to avoid completely empty regions. The number of tiles was constrained by labelling budget and so we report results using a k-fold biome-stratified cross-validation and a test (holdout) split. We partitioned 10% of the data for holdout testing, stratified such that the biome distribution in the train/test splits are approximately balanced. We then split the remaining 90% (train data) into 5 folds, also stratified by biome. There are no overlaps between source orthomosaics in the splits to avoid train/test leakage. To reduce file size, we distribute train and test splits as single MS-COCO formatted JSON files and provide the filenames/OAM IDs used for each split; a parallel release on HuggingFace contains the validation fold number in each Parquet entry and we provide a script to generate the training folds.

We also consider biome distribution as a measure of diversity. The WWF Terrestrial Ecoregions of the World [61] describe a hierarchical classification for regions on Earth that broadly align with common species distribution. The authors propose 867 ecoregions which are grouped into 14 biomes. In our dataset, we tag each tile with the biome index for the site, if it could be determined. This provides a coarse measure of the types of region which are over- or under-represented in our data. Some biomes, such as *(6) Boreal Forests/Taiga*, *(9) Flooded Grasslands and Savannahs* and *(11) Tundra* are not well-represented in the dataset due to limited availability and spatial biases of imagery in OAM, as well as the sampling technique used to select imagery. More information can be found in the supplementary material.

## 3.2 Image and label characteristics

We provide annotations for tiles of 2048 x 2048 pixels, uniformly resampled to 10 cm/px resolution. Individual tiles are random crops of "parent" surveys at higher than or equal to the target resolution. Each tile is a geo-referenced image in GeoTIFF format using the EPSG:3395 (World Mercator) projected coordinate reference system (CRS). Imagery is stored in 8-bit 3-channel RGB format with JPEG compression. 10 cm resolution was chosen as a trade-off between object visibility (ease of seeing trees and canopy structure) and ease of labelling with standard annotation tools. The large tile size allows for down-scaling to 0.8 m resolution at a tile size of 256 px, or up-scaling by re-sampling from parent imagery (when resolution permits). Thus, the dataset might also be used for assessing methods for satellite imagery analysis at approximately meter-scale resolution.

Labels are provided as semantic masks and in MS-COCO instance segmentation (polygon) format with two classes: `tree` and `canopy` (group of trees). This allows for different modeling approaches to be tested: individual tree and canopy region instance segmentation, or binary tree canopy semantic segmentation. An advantage of polygonal or semantic representations over bounding boxes is that predictions explicitly capture morphological information (i.e. crown shape) and makes area coverage estimation trivial. We did not consider acquisition time when sampling images which has an implication for deciduous trees which exhibit seasonal leaf cover variation. The dataset contains images of trees with and without leaves, and a few images that do not contain any annotations. The compressed dataset volume is 3.9 GB.

A key modeling decision for individual tree detection is whether to attempt to individually segment trees in regions of closed canopy (i.e. where trees are touching). Natural, established, forests have complex structure and it can be extremely difficult to delineate individuals in dense canopy, especially when multiple trees of the same species stand adjacent to each other. With other data like LIDAR or hyperspectral imaging, it is possible to exploit variations in canopy height, or leaf reflectance. Given the challenge of unambiguously segmenting individual trees in closed canopy and using RGB imagery alone, we chose to treat regions as a crowd class so that all regions are labelled, leaving the option open to re-assess annotations in the future to provide a more detailed segmentation.

Approaches in the literature vary. [78] provide bounding box annotations of trees derived from LIDAR surveys with a smaller set of human annotations. [18] segment trees within canopy, train a semantic segmentation model, and use a post-processing step to split detections. [11] do not attempt to annotate all trees, but do label individual instances in dense rainforest. [50] train a model with a semantic segmentation and head using a weighted loss function which penalises contact between segmented regions. Some of these works show compelling results from relatively few instance

annotations, O(1-10k), which implies it may be possible to selectively fully annotate images in OAM-TCD to achieve similar results.

### 3.3 Annotation process

Our annotation process started with weak labels generated by a Mask-RCNN model trained on a small portion of hand labelled data, followed by human annotation and review. This accelerated the labelling process in cases where the model performed well with little data, for example distinctive species like palms. We hired professional annotators that were not domain experts, but a majority of annotations were reviewed by a second person with ecology expertise. During the labelling process, there was a feedback loop to correct annotations and to answer questions from annotators.

Annotators were given comprehensive guidelines and we revised our process over time in order to address edge cases in labelling. We provide more information on our annotation guidelines in the supplementary material. Annotators were instructed to (1) label all trees in the image that they could confidently identify and (2) if it was not possible to identify individual trees within a group, mark it as "closed canopy". As a guideline, we suggested that if a region contained fewer than 5 connected trees, they should be annotated individually.

We explored multiple models for annotation pricing, including by overall annotation time, per-polygon and per-image. The sizes of object in our dataset vary from tens of pixels across to close to a full tile size (thousands of pixels) with diverse complexity. Annotation cost varied between 5-10 USD per image depending on the pricing structure and contractor; the cost of labelling the entire dataset was approximately 25k USD.

As our images are not sourced from ecological surveys, in almost all cases we do not have field ground truth available. We therefore aimed for conservative annotation - if it was not obvious whether a tree was an individual or multiple, we asked annotators to mark it as a canopy. However, when considering the labels as purely binary masks, we believe that our labels are generally self-consistent and are high quality.

### 3.4 Licensing and access

A key consideration for our dataset is permissive and open licensing. OAM declares that all imagery in their repository is licensed under Creative Commons Attribution 4.0 International (CC BY 4.0); however a subset of around 10% of the dataset images are labeled CC BY-NC 4.0 or CC BY-SA 4.0. We re-distribute image tiles under the same license as the metadata (as provided by OAM), which contains attribution information and links to the source orthomosaic. We split the dataset by common image license, so users may choose which combination is most appropriate for their application. In most cases, using only the CC BY imagery should yield good results; we reserve CC BY-SA imagery for testing only. For more information, see the supplementary material.

Our training and detection pipeline is hosted on GitHub (`https://github.com/Restor-Foundation/tcd` under an Apache 2.0 license. We provide a hosted version of the dataset on HuggingFace Hub in Apache Parquet format (`https://huggingface.co/restor/tcd`), as well as a mirror on Zenodo in MS-COCO format. The dataset is provided as "ML-ready" and can be used with off-the-shelf object detection frameworks with little to no modification.

## 4 Baseline Models

We trained exemplary models for instance and binary semantic segmentation and release them as usable products for the community. As there are a huge range of model architectures available, we limited training to representative examples of each task type. In order to assess model performance, we report metrics computed on the dataset (k-fold cross validation and holdout split); we also report quantitative and qualitative evaluations on independent data. There is limited like-for-like data available for third party testing so we constructed tests that demonstrate the strengths and weaknesses of the models. Our release models are trained on all training data, with metrics computed on the holdout set.

| | 5-fold Cross-Validation | | | Holdout | | |
|---|---|---|---|---|---|---|
| | IoU | Acc | F1 | IoU | Acc | F1 |
| UNet ResNet34 | 0.842±0.004 | 0.882±0.011 | 0.874±0.002 | 0.838 | 0.883 | 0.871 |
| UNet ResNet50 | 0.856±0.007 | 0.872±0.006 | 0.885±0.005 | 0.849 | 0.881 | 0.880 |
| SegFormer mit-b0 | 0.858±0.011 | 0.870±0.021 | 0.887±0.009 | 0.865 | 0.892 | 0.882 |
| SegFormer mit-b1 | 0.873±0.009 | 0.888±0.015 | 0.900±0.007 | 0.870 | **0.897** | 0.891 |
| SegFormer mit-b2 | 0.878±0.013 | 0.901±0.019 | 0.904±0.010 | 0.871 | 0.889 | 0.898 |
| SegFormer mit-b3 | 0.879±0.009 | 0.897±0.023 | 0.905±0.007 | 0.875 | 0.884 | 0.875 |
| SegFormer mit-b4 | 0.880±0.007 | 0.891±0.017 | 0.905±0.007 | 0.875 | 0.891 | 0.901 |
| SegFormer mit-b5 | **0.887±0.003** | **0.905±0.006** | **0.914±0.002** | **0.876** | 0.890 | **0.902** |

Table 1: Training results for semantic segmentation models, using the cross-validation and holdout splits in the OAM-TCD dataset. We report the mean and standard deviation of results for cross-validation folds.

Output examples of segmentation results from our models on independent data are shown in Figure 3 and discussed in the following section. A more detailed overview, including model cards, is included in the supplementary information and our GitHub repository.

**Dataset pre-processing and augmentation** All models were trained on random crops of 1024x1024 pixels, to provide a large spatial context of around 100 m$^2$ for predictions. We perform a series of random augmentation operations and apply normalisation using ImageNet statistics. The augmentation step includes: horizontal and vertical flipping, rotation, blur and colour adjustments (brightness, hue, saturation). Models are trained at a fixed spatial resolution of 10 cm/px and we do not perform tiled inference when testing on the holdout set. In all cases, we fine-tune models previously trained on MS-COCO or ImageNet which allowed faster convergence compared to models trained from scratch. A fixed seed of 42 was used for each training run for repeatability.

**Semantic segmentation** We trained the UNet architecture with ResNet34 and 50 backbones implemented by the Segmentation Models Pytorch library [40]. We also trained a series of SegFormer models [85], with size variants of mit-b0 through mit-b5. Pytorch Lightning [30] was used as our training backend and training logs (as Tensorboard event files) are released alongside the final models. Hyperparameter details may be found in the supplementary information.

Table 1 shows model performance from cross-validated models and on the holdout dataset; we report several binary metrics: accuracy, F1 and Jaccard Index (IoU) computed using the TorchMetrics library [25]. In general, SegFormer outperforms UNet, but there is limited gain from using larger model variants. Our hypothesis for the plateau in performance is that we are reaching the limits of the dataset in terms of annotation consistency and diversity, but it is encouraging that such good results can be obtained with relatively lightweight architectures. One advantage of UNet is that it is a well-understood architecture and is permissively licensed; SegFormer is released under a more restrictive NVIDIA research license but we expect similar results could be achieved using Pyramid Vision Transformers (PVT) v2 which shares many of the same architecture choices [76].

**Instance segmentation** For instance segmentation, we trained Mask-RCNN models with a ResNet50 backbone, using the Detectron2 framework [82] (`restor/tcd-mask-rcnn-r50` on HuggingFace). Models were trained with mostly default hyperparameters, with a fixed number of iterations (100,000) at batch size 8, taking the best periodic checkpoint as the final version. We used a tuned learning rate of 0.001 with a stepped schedule following the original paper: a learning rate multiplier of 0.1 applied at iterations 80000 and 90000. Due to high object density in the images, we increased the number of predictions to 512. Cross-validation performance of the models is mAP50=41.79±1.38, and mAP50=43.22 on the holdout set using the COCO API `segm` task. For comparison, DeepForest report baseline results of mAP50=50 on NeonTreeEvaluation albeit on bounding boxes and not instance predictions; DeepForest also attempts to predict all trees individually.

There are several transformer-based successors to Mask-RCNN, from the DETR [20] family of models to Mask2Former [23]. While we experimented with these architectures, and provide sample

training code in the repository, we were unable to reliably outperform Mask-RCNN and found poorer convergence behaviour. Additionally we found there were practical limitations with RAM usage when predicting large numbers of objects, due to implementation/mask representation details. Since tree crowns are - to first order - simple, convex and non-overlapping shapes (deformed circles), there may be value in exploring other memory-efficient architectures that explicitly predict polygon representations, such as DPPD [87] or Poly-YOLO [39].

**Environmental impact**  Our release models were all trained on a single NVIDIA RTX3090 GPU with 24 GB of VRAM at a reduced power target of 200 W. Additional experiments and dataset generation were performed on Google Cloud Platform and ETH's Euler compute cluster. The largest models took approximately 1 day to train. Using the MachineLearning Impact calculator [46], we estimate 3.63 kg $CO_2$ equivalent per model variant. This does not take into account testing and failed model runs and so underestimates the overall emission cost of the project, however this can be used as a guideline for emissions generated when training further models.

## 5    Third-party benchmarks

To assess our models on out-of-domain data, and for practical applications, we report results on two independent datasets. For brevity, we report results from the best models here. Examples of predictions from test data are shown in Figure 3. Geo-spatial data often far exceeds the VRAM capacity of even a large GPU and images must be processed in a tiled fashion; broadly, we follow the recommendations in [38] for semantic segmentation and we use heuristics to merge overlapping instances (described in the supplementary material).

**Urban tree detection - Zurich**  The Federal Institute of Topography in Switzerland (Swisstopo) produces 3-year aerial surveys of the entire country, at 10 cm resolution. By combining 2022 imagery with a up-to-date municipal tree inventory from the city of Zurich, we can assess model performance against an accurate ground truth. A LIDAR-derived CHM within the city boundary, an area of approximately 90 km$^2$, is also available. The inventory data also provides tree species and crown diameter for almost 77k trees within the city. We exclude trees which are marked as planted in 2022, as the inventory and survey dates are not coincident, leaving around 71k keypoints.

We produced prediction maps for canopy coverage and individual trees for the city of Zurich. Our segmentation models show excellent agreement with the LIDAR CHM (for `tcd-segformer-mit-b5`: IoU=0.791, Accuracy=0.922, F1=0.883) and in some cases provide superior delineation of canopy due to the lower resolution of LIDAR and potentially inaccurate point cloud classification (e.g. returns from some street lamps are present in the CHM). This suggests that our segmentation model could be used in a complementary fashion to better classify LIDAR-derived height rasters, particularly in urban environments. Another example of semantic segmentation output is shown in Figure **??** and a full image of the city can be found in the supplementary material (Figure **??** in this document).

Results from instance segmentation also show good agreement with the municipal tree inventory. Since our models detect non-municipal trees, it is not possible to measure the false positive rate. We report a recall of 0.36 for keypoints that were matched to tree labels and 0.73 when also including canopy regions. Filtering predictions using a semantic segmentation map appears to be an effective way of removing high confidence false positives like tree shadow. The relatively low recall for keypoint-in-tree predictions is likely because many trees in the city are close together and are predicted as groups. Our model identifies 193k individual trees within the prediction boundary, the vast majority of which are not actively monitored.

**WeRobotics Open AI Challenge**  This dataset is a tree detection benchmark. The input is a 325 ha orthomosaic captured over the Kingdom of Tonga [3], hosted on OAM. Tree centers for four species were annotated by humans (13402 total keypoints); not all trees in the image are labelled and our model predicts around twice this many instances over the orthomosaic. Similar to our analysis of Zurich for instance segmentation, we check whether trees in the dataset are captured by either tree or canopy polygons from our model output. We report a recall of 0.64 for keypoints that were matched to tree labels and 0.94 when also including canopy regions. Prediction results are shown in Figure 3 for instance and semantic segmentation output.

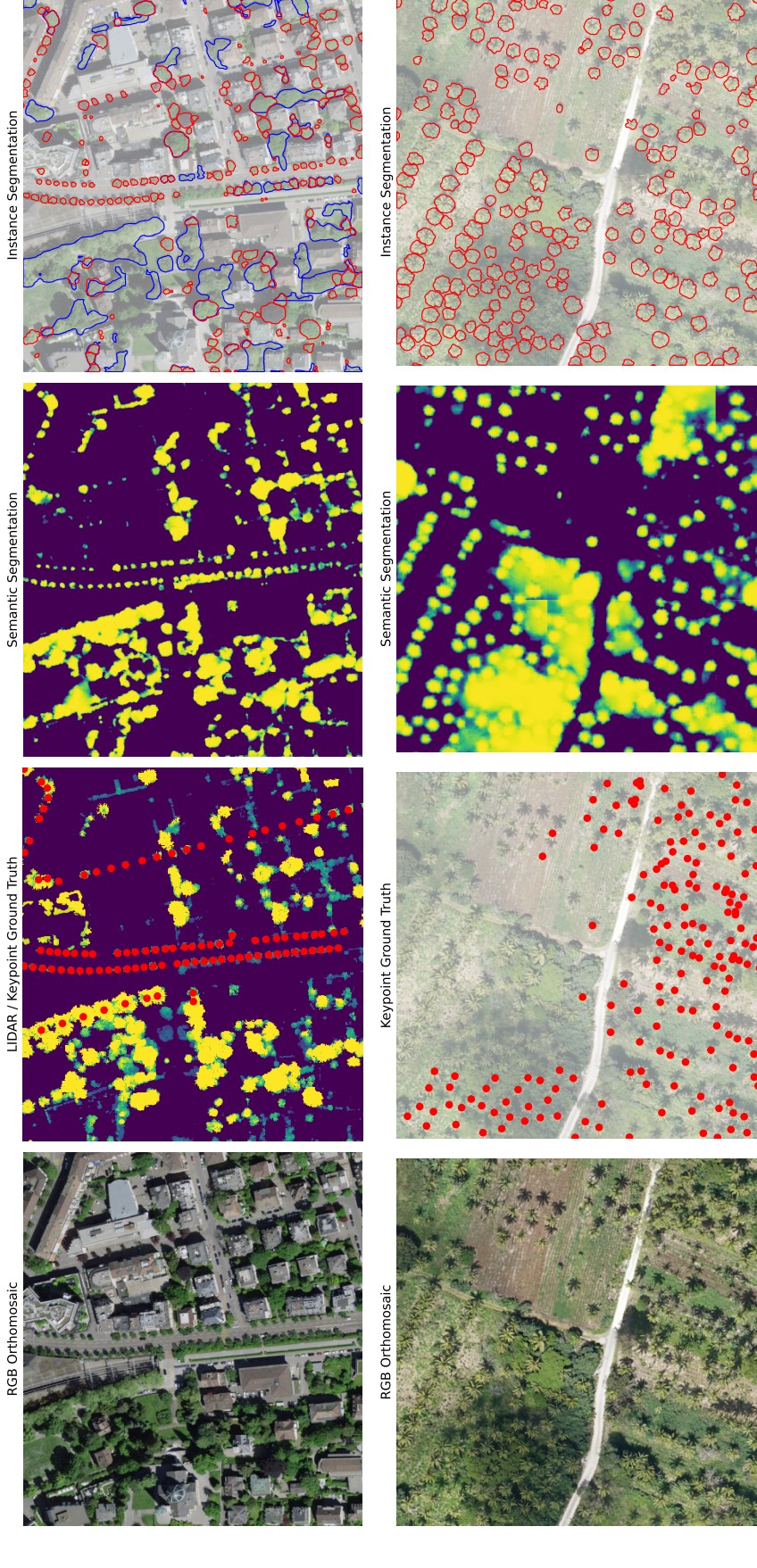

Figure 3: Model predictions for independent benchmark data. Upper series shows predictions over Zurich in reference to a ground truth canopy height model and tree keypoints, lower series shows predictions on a large orthomosaic from the Kingdom of Tonga with ground truth tree keypoints. Red instances are predicted trees, blue instances are predicted groups. In both datasets only some trees have ground truth labels. Most of the missing detections in the Tonga example above are from small Banana (*Musa*) plants. Models used are restor/tcd-segformer-mit-b5 (threshold 0.5) and restor/mask-rcnn-r50 (threshold 0.4). Plotting code may be found in the pipeline repository.

# 6 Ethics, Limitations and Impact

We hope that the release of this dataset will bring a positive improvement to the ecological monitoring landscape. There are three main limitations that we highlight here. First, we do not provide extensive validations on in-situ field data with this paper and users should not use these models for decision-making purposes without additional assessments on their own data. Second, there are inconsistencies in the human labelling process and while we have tried to address this by double-reviewing a subset of the imagery, there is noise and bias in the dataset. Some biomes are under-represented, as are some geographic regions. Finally, we did not target annotating every tree in our dataset images. Our instance segmentation models are unlikely to perform well in environments with large regions of dense canopy which users may expect. The model and dataset cards provide more information on these limitations and biases.

Viewing our work from an ethical perspective, we ask: what advantage do our models give to malicious or unethical users over possession of very high resolution imagery alone? For applications that involve land or resource exploitation, other lower resolution data may be sufficient to identify valuable targets and our dataset does not include species information. A tangential result of our work is that we provide a concrete data point for cost:performance which may encourage users to replicate our work (or expand upon it) for unethical use cases. On the other hand, providing open-source mapping tools that can be used with low-cost UAV and aerial images is important for accessibility in areas where satellite coverage is limited or expensive. Our aim is to facilitate tree mapping at a higher resolution than is currently possible, ultimately to improve transparency in the conservation and ecology communities.

**PII** We include links to metadata with imagery which may link to personally identifiable information (e.g. names and email addresses for rights holder information) for each image, however this is already in the public domain on OAM, provided by the image uploader. We make the assumption that imagery provided through OAM has the permission of the copyright holder and we provide the metadata for attribution purposes. The images themselves are too low resolution to contain PII (for example human faces).

# 7 Conclusion and further work

There are several avenues for future work: we hope to improve biome coverage in the dataset by including more imagery from under-represented regions; there is ongoing work to improve the consistency of the annotations, for example identifying group labels that could be split into individuals. We are facilitating an ongoing citizen science campaign [9] which aims to provide an assessment of inter-annotator agreement and confidence for individual labels. Some of these improvements could also be posed as challenges - for example identifying mislabeled objects.

On its own, we believe that OAM-TCD is the largest open dataset of its kind and will provide other researchers with a novel benchmark for high resolution tree detection, for example zero-shot evaluation of foundational Earth Observation (EO) models. We release our prediction pipeline as a user-friendly tool for tree mapping and hope it will be of use to the ecological community.

## Acknowledgements

This project was developed through a collaboration between the Restor Foundation and the DS3Lab at ETH Zurich. Funding for research staff and data labelling was supported by a Google.org AI for Social Good grant (ID: TF2012-096892, *AI and ML for advancing the monitoring of Forest Restoration*).

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
