# OAM-TCD Appendix / Supplementary Material

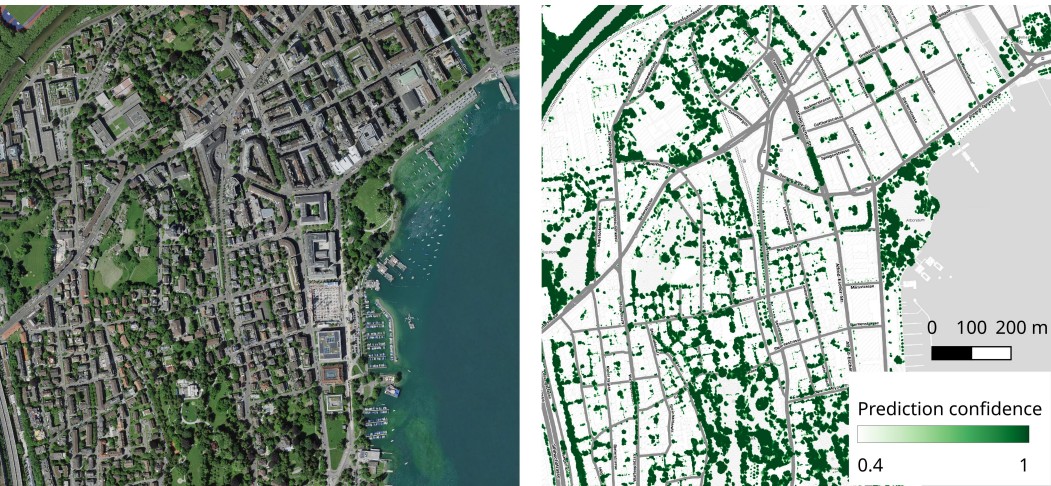

Figure 1: Tree semantic segmentation for Zurich, predicted at 10 cm/px. Predictions with a confidence of < 0.4 are hidden. Left: 10 cm RGB orthomosaic provided by the Swiss Federal Office of Topography swisstopo/SWISSIMAGE 10 cm (2022), Right: prediction heat map. Zooming in is recommended to see small details, e.g. trees along the top edge of the lake. Base map tiles by Stamen Design, under CC BY 4.0. Data by OpenStreetMap, under ODbL.

## 1 Dataset Information

### 1.1 Dataset Card

This dataset card is reproduced from our HuggingFace repository (`https://huggingface.co/datasets/restor/tcd`), using the provided template as reference for headings. Minor changes have been made for readability and formatting.

**Dataset Details**   OAM-TCD is a dataset of high-resolution (10 cm/px) tree cover maps with instance-level masks for 280k trees and 56k tree groups. Images in the dataset are provided as 2048x2048 px RGB GeoTIFF tiles. The dataset can be used to train both instance segmentation models and semantic segmentation models.

**Dataset Description**

- Curated by: Restor and ETH Zurich
- Funded by: Restor and ETH Zurich, supported by a Google.org AI for Social Good grant (ID: TF2012-096892, AI and ML for advancing the monitoring of Forest Restoration)
- License: Annotations are predominantly released under a CC BY 4.0 license, with around 10% licensed as CC BY-NC 4.0 or CC BY-SA 4.0. These less permissive images are distributed in separate repositories to avoid any ambiguity for downstream use.

**Dataset Sources**   All imagery in the dataset is sourced from OpenAerialMap (OAM, part of the Open Imagery Network / OIN).

**Dataset License**   OIN declares that all imagery contained within is licensed as CC BY 4.0 (`https://github.com/openimagerynetwork/oin-register`) however some images are labelled as CC BY-NC 4.0 or CC BY-SA 4.0 in their metadata.

To ensure that image providers' rights are upheld, we split these images into license-specific repositories, allowing users to pick which combinations of compatible licenses are appropriate for their application. We have initially released model variants that are trained on CC BY 4.0 + CC BY-NC

4.0 imagery. CC BY-SA 4.0 imagery was removed from the training split, but it can be used for evaluation.

## Uses

We anticipate that most users of the dataset wish to map tree cover in aerial orthomosaics, either captured by drones/unmanned aerial vehicles (UAVs) or from aerial surveys such as those provided by governmental organisations.

**Direct Use**   The dataset supports applications where the user provides an RGB input image and expects a tree (canopy) map as an output. Depending on the type of trained model, the result could be a binary segmentation mask or a list of detected trees/groups of tree instances. The dataset can also be combined with other license-compatible data sources to train models, aside from our baseline releases. The dataset can also act as a benchmark for other tree detection models; we specify a test split which users can evaluate against, but currently there is no formal infrastructure or a leader board for this.

**Out-of-Scope Use**   The dataset does not contain detailed annotations for trees that are in closed canopy i.e. are touching. Thus the current release is not suitable for training models to delineate individual trees in closed canopy forest. The dataset contains images at a fixed resolution of 10 cm/px. Models trained on this dataset at nominal resolution may under-perform if applied to images with significantly different resolutions (e.g. satellite imagery).

The dataset does not directly support applications related to carbon sequestration measurement (e.g. carbon credit verification) or above ground biomass estimation as it does not contain any structural or species information which is required for accurate allometric calculations (16). Similarly models trained on the dataset should not be used for any decision-making or policy applications without further validation on appropriate data, particularly if being tested in locations that are under-represented in the dataset. See Table 1 in this document.

**Dataset Structure**   The dataset contains pairs of images, semantic masks and object segments (instance polygons). The masks contain instance-level annotations for (1) individual trees and (2) groups of trees, which we label `canopy`. For training our models we binarise the masks. Metadata from OAM for each image is provided and described in Section 1.2. Example annotations are shown in Figure 2.

The dataset is released with suggested training and test splits, stratified by biome. These splits were used to derive results presented in the main paper. Where known, each image is also tagged with its terrestrial biome index [-1, 14]. This relationship was defined by looking for intersections between tile polygons and reference biome polygons, an index of -1 means a biome wasn't able to be matched. Tiles sourced from a given OAM image are isolated to a single fold (and split) to avoid train/test leakage.

k-fold cross-validation indices within the training set are also provided ($k = 5$). That is, each image is assigned an integer [0, 4] which refers to a validation fold. Users are also free to pick their own validation protocol (for example one could split the data into biome folds), but results may not be directly comparable with results from the release paper.

## Dataset Creation

**Curation Rationale**   The use-case within Restor (2) is to feed into a broader framework for restoration site assessment. Many users of the Restor platform are stakeholders in restoration projects; some have access to tools like UAVs and are interested in providing data for site monitoring. Our goal was to facilitate training tree canopy detection models that would work robustly in any location. The dataset was curated with this diversity challenge in mind - it contains images from around the world and (by serendipity) covers most terrestrial biome classes.

It was important during the curation process that the data sources be open-access and so we selected OpenAerialMap as our image source. OAM contains a large amount of permissively licensed global imagery at high resolution (chosen to be < 10 cm/px for our application).

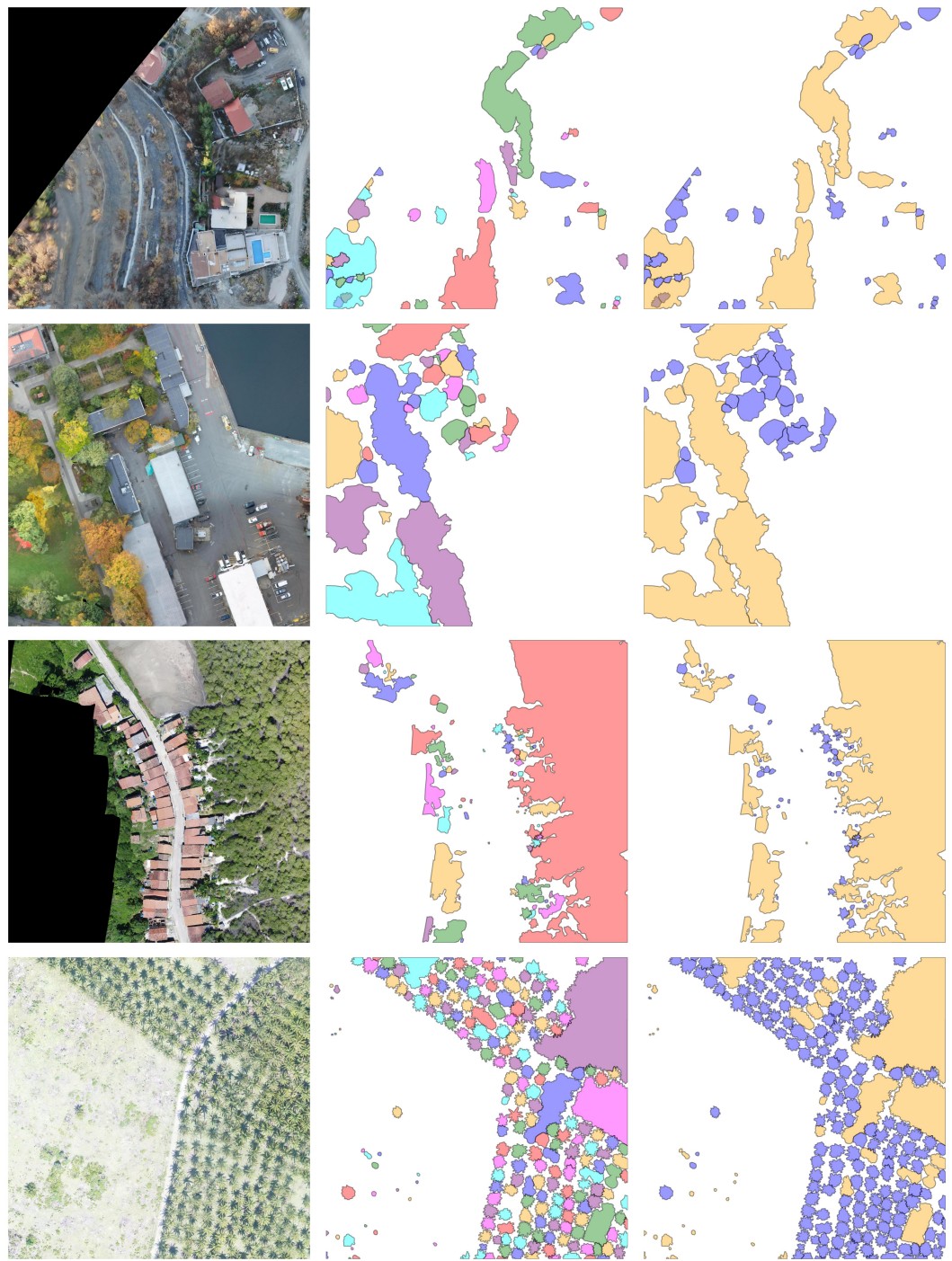

Figure 2: Further example annotations from the OAM-TCD test split. Left: RGB image, Middle: ground truth segmentation randomly coloured by segment ID, Right: coloured by class - blue = tree, orange = tree canopy. All images licensed CC BY 4.0, contributors to Open Imagery Network, top-bottom OAM-TCD IDs: 555,1445,1594,2242.

**Source Data**

**Data Collection and Processing**  We used the OAM API to download a list of surveys on the platform. Using the metadata, we discarded surveys that had a ground sample distance of greater than 10 cm/px (for example satellite imagery). The remaining sites were binned into 1 degree square regions across the world. There are sites in OAM that have been uploaded as multiple assets, and naive random sampling would tend to pick several from the same location. We then sampled sites from each bin and random non-empty tiles from each site until we had reached around 5000 tiles. This was arbitrarily constrained by our estimated annotation budget.

Interestingly we did not make any attempt to filter for images containing trees, but in practice there are few negative images in the dataset. Similarly we did not try to filter for images captured in a particular season, so there are trees without leaves in the dataset.

**Who are the source data producers?**  The images are provided by users of OpenAerialMap / contributors of Open Imagery Network.

**Annotation**

**Annotation process**  Annotation was outsourced to commercial data labelling companies who provided access to teams of professional annotators. We experimented with several labelling providers and compensation strategies.

Annotators were provided with a guideline document that provided examples of how we expected images should be labeled. This document evolved over the course of the project as we encountered edge cases and questions from annotation teams. As described in the main paper, annotators were instructed to attempt to label open canopy trees individually (i.e. trees that were not touching). If possible, small groups of trees should also be labelled individually and we suggested < 5 trees as an upper bound. Annotators were encouraged to look for cues that indicated whether an object was a tree or not, such as the presence of (relatively long) shadows and crown shyness (inter-crown spacing). Larger groups of trees, or ambiguous regions would be labelled as "canopy". Annotators were provided with full size image tiles (2048 x 2048) and most images were annotated by a single person from a team of several annotators.

There are numerous structures for annotator compensation - for example, paying per polygon, paying per image and paying by total annotation time. The images in OAM-TCD are complex and a fixed cost per image was excluded early on as the reported annotation time varied significantly. Anecdotally we found that the most practical compensation structure was to pay for a fixed block of annotation time with regular review meetings with labeling team managers. Overall, the cost per image was between 5-10 USD and the total annotation cost was approximately 25k USD. The labelling companies that we contracted declared that they compensated annotators fairly within their country of domicile. Unfortunately we do not have accurate estimates for time spent annotating all images, but we did advise annotators that if they spent more than 45-60 minutes on a single image that they should flag it for review.

**Who are the annotators?**  We did not have direct contact with any annotators and their identities were anonymised during communication, for example when providing feedback through managers.

**Bias and Risks**  There are several potential sources of bias in our dataset. The first is geographic, related to where users of OAM are likely to capture data - accessible locations that are amenable to UAV flights. Some locations and countries place strong restrictions on UAV possession and use, for example. One of the use-cases for OAM is providing traceable imagery for OpenStreetMap which is also likely to bias what sorts of scenes users capture. The sites in OAM-TCD are most likely not restoration projects and we did not find any overlap between sites in OAM and Restor at the time of sampling.

The second is bias from annotators, who were not ecologists. Benchmark results from models trained on the dataset suggest that overall label quality is sufficient for accurate semantic segmentation. However, for instance segmentation annotators had freedom the choose whether to individually label trees or not. This naturally resulted in some inconsistency between what annotators determined was a

tree, and at what point to annotate a group of trees as a group. In the main paper, we discuss the issue of conflicting definitions for "tree" among researchers and monitoring protocols.

Figure 2 highlights some of the inconsistencies described above. Some annotators labeled individual trees within group labels; in the bottom plot most palm trees are individually segmented, but some groups are not. A goal for the project is to attempt to improve label consistency, identify incorrect labels and attempt to split group labels into individuals. After annotation was complete, we contracted two different labelling organisations to review (and re-label) subsets of the data; we have not released this data yet, but plan to in the future.

Biases related to biome coverage are discussed in section 1.3 and in Table 1.

The greatest risk that we foresee om releasing this dataset is usage in out-of-scope scenarios. For example, using trained models on imagery from regions/biomes that the dataset is not representative of without additional validation. Similarly there is a risk that users apply the model in inappropriate ways, such as measuring canopy cover on imagery taken during periods of abscission (when trees lose leaves). It is important that users carefully consider timing (seasonality) when comparing time-series predictions.

While we believe that the risk of malicious or unethical use is low - given that other global tree maps exist and are readily available - it is possible that models trained on the dataset could be used to identify areas of tree cover for illegal logging or other forms of land exploitation. Given that our models can segment tree cover at high resolution, it could also be used for automated surveillance or military mapping purposes.

**Personally identifiable information**  Contact information is present in the metadata for imagery. We do not distribute this data directly, but each image tile is accompanied by a URL pointing to a JSON document on OpenAerialMap where it is publicly available. Otherwise, the imagery is provided at a low enough resolution that it is not possible to identify individual people.

The image tiles in the dataset contain geospatial information which is not obfuscated, however as one of the purposes of OpenAerialMap is humanitarian mapping (e.g. tracing objects for inclusion in OpenStreetMap), accurate location information is required and uploaders are informed that this information would be available to other users. We also assume that image providers had the right to capture imagery where they did, including following local regulations that govern UAV activity.

An argument for retaining accurate geospatial information is that annotations can be verified against independent sources, for example global land cover maps. This allows for combination with other geo-referenced datasets like multispectral satellite imagery or products like Global Ecosystem Dynamics Investigation (GEDI) data (3).

Contact information is provided in our repository that discusses how users/rights holders can request for imagery to be removed.

## 1.2   Image Metadata

We include metadata for the source image of each tile. Some key fields are replicated in the dataset, the full metadata is provided as a URL. The metadata follows the Open Imagery Network specification, described at `https://github.com/openimagerynetwork/oin-metadata-spec`. Briefly, this includes the bounding coordinates of the image, its coordinate reference system, ground sample distance, contact information for the image supplier and information about the acquisition (dates, aerial platform, etc.).

## 1.3   General Dataset Statistics

The dataset contains 5072 image tiles sourced from OpenAerialMap; of these 4608 are licensed as CC BY 4.0, 272 are licensed as CC BY-NC 4.0 and 192 are licensed as CC BY-SA 4.0. As described earlier, we split these images into separate repositories to keep licensing distinct. Only around 5% of imagery in the training split has a less permissive non-commercial license and we are re-training models on only the CC BY portion of the data to maximise accessibility and re-use.

The training dataset split contains 4406 images and the test split contains 666 images. All images are the same size (2048x2048 px) and the same ground sample distance (10 cm/px). The geographic distribution of the dataset is shown in the main paper.

**Biome Distribution**  Table 1 shows the number of tiles that correspond to each of the 14 terrestrial biomes described by Olson et. al (14). The majority of the dataset covers (1) tropical and temperate broadleaf forest. Some biomes are clearly under-represented - notably (6) boreal forest/taiga; (9) flooded grasslands and savannas; (11) tundra; and (14) mangrove. Some of these biomes, mangrove in particular, are likely under-represented due to our sampling method (by binned location), as their geographic extent is relatively small. These statistics could be used to guide subsequent data collection in a more targeted fashion.

It is important to note that the biome classification is purely spatial and without inspecting images individually, one cannot make assumptions about what type of landscape was actually imaged, or if it is a natural ecosystem representative of that biome. We do not currently annotate images with a land use category, but this would potentially be a useful secondary measure of diversity in the dataset.

**Dataset splits**  Since the dataset is relatively small - just over 5000 images - we opted to perform a 5-fold cross validation to better estimate model performance and to allow for training on more data at release time.  Folds are stratified over terrestrial biomes using the `model_selection.StratifiedKFold` function in `scikit-learn` (15). Table 1 also shows statistics for the cross-validation folds (e.g. fold size). Since we stratify at the level of source imagery index (`oam_id`)to avoid leaking tiles between folds, there is some variation in fold size due to differing numbers of tiles from each source image.

In addition to cross-validation, approximately 10% of the images (plus all CC BY-SA 4.0 images) are reserved as a test/holdout set which was not used during hyperparameter tuning and model experimentation. This set is only used to evaluate our final "release" models.

As described in the dataset card, validation fold IDs are assigned to each image and filter operations can be used on the dataset table to construct appropriate subsets for training.

## 1.4  Hosting and Access

Our dataset is hosted on two platforms for better availability and to mitigate against host failures:

- DOI: `10.5281/zenodo.11617167`

- HuggingFace Hub: `https://huggingface.co/datasets/restor/tcd`

- Zenodo: `https://zenodo.org/records/11617167`

The release on HuggingFace is provided in Apache Parquet format and can be downloaded using the HuggingFace `datasets` library. The repository contains images, masks and metadata - including MS-COCO annotation records for each image. The annotation format is technically MS-COCO panoptic (mask images contain RGB-encoded instance IDs), though we do not label all pixels. Labels can be used with many of the segmentation models on the `transformers` platform with minor modification.

Many existing object and instance detection frameworks support MS-COCO format annotations, for example Detectron2 (which we use) and the mmdetection(1) ecosystem. To more easily support these frameworks, and also to provide an alternative hosting platform, the dataset on Zenodo is provided in MS-COCO format.

We provide code in our repository (`tools/generate_dataset.py`) to generate MS-COCO annotation files for each split and fold. This tool also supports combining datasets. This script was used to generate the release files uploaded to Zenodo.

In the near future we aim to provide a Spatio-Temporal Asset Catalog (STAC) with the dataset to allow standardised querying and access.

| Cross-validation fold | 1 | 2 | 3 | 4 | 5 | Holdout | Total |
|---|---|---|---|---|---|---|---|
| **Biome** | | | | | | | |
| **Unmatched** | 46 | 18 | 34 | 39 | 20 | 59 | 206 |
| **(1) Tropical & Subtropical Moist Broadleaf Forests** | 268 | 412 | 378 | 377 | 295 | 276 | 2006 |
| **(2) Tropical & Subtropical Dry Broadleaf Forests** | 53 | 68 | 12 | 48 | 46 | 13 | 240 |
| **(3) Tropical & Subtropical Coniferous Forests** | 11 | 6 | 13 | 19 | 10 | 6 | 65 |
| **(4) Temperate Broadleaf & Mixed Forests** | 354 | 368 | 260 | 371 | 277 | 202 | 1832 |
| **(5) Temperate Coniferous Forests** | 74 | 43 | 51 | 46 | 28 | 55 | 297 |
| **(6) Boreal Forests/Taiga** | 0 | 0 | 0 | 8 | 0 | 3 | 11 |
| **(7) Tropical & Subtropical Grasslands, Savannas, and Shrublands** | 15 | 34 | 20 | 28 | 37 | 8 | 142 |
| **(8) Temperate Grasslands, Savannas, & Shrublands** | 4 | 5 | 14 | 15 | 12 | 34 | 84 |
| **(9) Flooded Grasslands & Savannas** | 0 | 10 | 0 | 0 | 0 | 0 | 10 |
| **(10) Montane Grasslands & Shrublands** | 0 | 6 | 12 | 0 | 0 | 0 | 18 |
| **(11) Tundra** | 0 | 9 | 0 | 0 | 0 | 0 | 9 |
| **(12) Mediterranean Forests, Woodlands, & Scrub** | 12 | 11 | 27 | 16 | 12 | 2 | 80 |
| **(13) Deserts & Xeric Shrublands** | 1 | 13 | 6 | 1 | 6 | 7 | 34 |
| **(14) Mangrove** | 12 | 0 | 2 | 1 | 12 | 1 | 28 |
| **Validation image tiles** | 850 | 1003 | 829 | 969 | 755 | 666 | 5072 |

Table 1: Number of image tiles in OAM-TCD for each terrestrial biome class, for each cross-validation fold in the dataset. "Unmatched" tiles were unable to be matched to a biome via polygon intersection. Also shown is the number of tiles per fold, and the distribution of biomes throughout the entire dataset.

## 1.5 Croissant

The dataset, as hosted on HuggingFace, contains an automatically generated Croissant metadata record: `https://huggingface.co/api/datasets/restor/tcd/croissant`.

## 2 Models

We release the following models on HuggingFace. SegFormer models can be downloaded directly using the `transformers` library, other model weights must be used in conjunction with other libraries or are automatically downloaded with our inference pipeline.

- SegFormer mit-b0: `restor/tcd-segformer-mit-b0`

- SegFormer mit-b1: `restor/tcd-segformer-mit-b1`

- SegFormer mit-b2: `restor/tcd-segformer-mit-b2`

- SegFormer mit-b3: `restor/tcd-segformer-mit-b3`

- SegFormer mit-b4: `restor/tcd-segformer-mit-b4`

- SegFormer mit-b5: `restor/tcd-segformer-mit-b5`

- UNet ResNet34: `restor/tcd-unet-r34`

- UNet ResNet50: `restor/tcd-unet-r50`

- Mask-RCNN ResNet50: `restor/mask-rcnn-r50`

## 2.1 Model Card(s)

This model card is reproduced from our HuggingFace repository, using the provided template as reference for headings. Minor changes have been made for readability and formatting, and for the sake of brevity we have combined cards for instance and segmentation models, highlighting differences where appropriate. There is some duplication of information between model and dataset cards.

- Developed by: Restor and ETH Zurich

- Funding: Restor and ETH Zurich, supported by a Google.org AI for Social Good grant (ID: TF2012-096892, AI and ML for advancing the monitoring of Forest Restoration)

- License: Mask-RCNN and UNet weights are currently licensed as CC BY-NC 4.0. Seg-Former weights are licensed under the NVIDIA Source Code License for SegFormer[1].

**Semantic Segmentation** models were trained on global aerial imagery from OAM-TCD and are able to accurately delineate tree cover in similar images. The models do not detect individual trees, but provide a per-pixel binary classification (tree/not tree). Post-processing such as connected components could be used to convert results into instance polygons.

**Instance Segmentation** models can predict individual trees in open canopy environments. Model output is instance polygons for trees and groups of trees.

**Model Sources** References for each model type may be found here: (10; 17; 18). Training code for all models is provided in our repository: `https://github.com/restor-foundation/tcd` and more information may be found in the accompanying main paper.

### Uses

The primary use-case for these model is assessing canopy cover from aerial images (i.e. percentage of study area that is covered by tree canopy).

**Direct Use** Models on their own are suitable for inference on a single image tile (typically 2048 px or smaller). For performing predictions on large orthomosaics, a higher level framework is required to manage tiling source imagery and stitching predictions. Our repository provides a comprehensive reference implementation of such a pipeline and has been tested on extremely large images (country-scale).

The model returns predictions for an entire image. In most cases users will want to predict cover for a specific region of the image, for example a study plot or some other geographic boundary. Some kind of region-of-interest analysis on the results is therefore required. Our linked pipeline repository supports standard shapefile-based region analysis and evaluation.

Direct model usage with the `transformers` library is straightforward for SegFormers:

```
from transformers import AutoImageProcessor
from transformers import AutoModelForSemanticSegmentation
import torch

processor = AutoImageProcessor.from_pretrained("restor/tcd-segformer-mit-b0")
model = AutoModelForSemanticSegmentation.from_pretrained("restor/tcd-segformer-
    mit-b0")

x = torch.randint(255, (3,512,512))
inputs = processor(images=[x], return_tensors="pt")
preds = model(pixel_values=inputs.pixel_values)
```

---

[1]See `https://github.com/NVlabs/SegFormer/blob/master/LICENSE`, accessed 1 June 2024.

**Out-of-Scope Use**    While we trained models on globally diverse imagery, some ecological biomes are under-represented in the training dataset. We therefore encourage users to experiment with their own imagery before using the model for mission-critical use.

Models were trained on imagery at a resolution of 10 cm/px and predictions on other resolutions may be unreliable. We recommend fine-tuning on images with alternative resolutions if that is required.

The model does not predict biomass, canopy height or other derived information. It only predicts the likelihood that some pixel (or polygon) is a tree (canopy). As-is, the model is not suitable for carbon estimation or similar activities.

## Bias, Risks, and Limitations

The main limitation of these models is false positive predictions over objects that look like, or could be confused as, trees. For example large bushes, shrubs or ground cover that looks like tree canopy. This is particularly a concern if images with resolutions significantly different to 10 cm/px. Some conservation/monitoring protocols specify height bounds to determine which objects are considered trees or not; our model does not predict this, but it could be used in conjunction with a LIDAR or photogrammetry surface model to filter appropriate points.

The dataset used to train this model was annotated by non-experts. We believe that this is a reasonable trade-off given the size of the dataset and the results on independent test data, as well as empirical evaluation during experimental use at Restor on partner data. However, there are almost certainly incorrect or inconsistent labels in the dataset and this may translate into incorrect predictions or other biases in model output. We have observed that semantic segmentation models tend to "disagree" with training data in a way that is reasonable (i.e. the aggregate statistics of the labels are good) and we are working to re-evaluate all training data to remove spurious labels. As we note in the main paper, there is no uniform definition for what a tree is, so it is impossible to construct a dataset that satisfies all use-cases unless it also contains per-instance species annotations.

We provide cross-validation results to give a robust estimate of prediction performance, as well as results on independent imagery (i.e. images the model has never seen) so users can make their own assessments. We do not provide any guarantees of accuracy and users should perform their own independent testing prior to deployment.

## Training Data

Models were trained using fine-tuning/transfer learning from ImageNet and MS-COCO. Otherwise the only training data used was OAM-TCD. See section 1.

## Training Procedure

We used a 5-fold cross-validation process to tune hyperparameters, before training on the whole training set and evaluating on the OAM-TCD holdout set. The model checkpoints in the `main` branches of the repository should be considered release versions; this is the default branch used when downloading from HuggingFace Hub.

**Semantic Segmentation**    Pytorch Lightning was used to train semantic segmentation models.

**Instance Segmentation**    The Detectron2 framework was used to train Mask-RCNN-based models.

**Preprocessing**    For SegFormer models, we use the provided pre-processor class that performs normalisation (ImageNet statistics) and converts the input to a Pytorch tensor. We do not resize input images (so that the geospatial scale of the source image is respected) and we assume that normalisation is performed in this processing step and not as a dataset transform. A similar approach was followed to train UNet and Detectron2-based models.

**Semantic segmentation training hyperparameters**    shown in Table 2

| | |
|---|---|
| **Image size** | 1024 x 1024 px |
| **Base learning rate** | 1e-4 to 1e-5 |
| **Scheduler** | Reduce on plateau |
| **Optimizer** | AdamW |
| **Batch size** | 32 |
| **Augmentation** | Random crop of source image to 1024x1024, arbitrary rotation, horizontal and vertical flips, colour adjustments |
| **Epochs** | 75 for cross-validation to ensure convergence; 50 for final models |
| **Normalisation** | ImageNet statistics |
| **Loss function** | Focal loss (UNet), Cross Entropy (SegFormer) |

Table 2: Hyperparameters used for semantic segmentation models.

| | |
|---|---|
| **Image size** | 1024 x 1024 px |
| **Base learning rate** | 1e-3 |
| **Schedule** | Stepped; Reduced 10x at 80% and 90% of training |
| **Optimizer** | AdamW |
| **Batch size** | 8 |
| **Augmentation** | Random crop of source image to 1024x1024, arbitrary rotation, horizontal and vertical flips, colour adjustments |
| **Iterations** | 100,000 |
| **Normalisation** | ImageNet statistics |
| **Loss function** | Mask R-CNN Loss |

Table 3: Hyperparameters used for instance segmentation (Mask-RCNN) models

**Instance segmentation hyperparameters**    shown in Table 3; the training schedule is largely the default suggested in Detectron2, with adjustments to the learning rate, batch size and number of fine-tuning iterations.

**Training curves**    Training curves can be found as Tensorboard records in the model repositories; HuggingFace supports a built-in Tensorboard instance for viewing online. Selected training metrics are shown in Figures 3, 4 and  5: aggregated training and validation losses, accuracy, F1 and IoU/Jaccard Index; y-axis labels indicate the metric label used during training for reference against logs. Aggregate curves show the mean and standard deviation bounds for metrics tracked in cross-validation. For Mask-RCNN, we use Detectron2's default logged metrics and report mAP and mAP50 (segmentation). The periodic spikes in validation loss for UNet models appear to be an artifact as overall validation performance is smoothly varying.

**Training $CO_2$ emissions estimate**

Table 4 shows $CO_2$ emissions estimates for model training. We assume the default carbon efficiency value (0.432 kg/kWh) given by Lacoste et al. (13). GPU: NVIDIA RTX 3090, power target 200 W. System: AMD Ryzen 7950X (TDP 170 W). CPU + peripherals estimated usage at 150W during training.

**Inference and training speed**

Models can be evaluated on CPU, but this requires a large amount of RAM for large tile sizes. In general we find that 1024 px inputs are a sensible upper limit, given the fixed size of the output segmentation masks in SegFormer models (i.e. it is probably better to perform inference in batched mode at 1024x1024 px than try to predict a single 2048x2048 px image).

All models were trained on a single GPU with 24 GB VRAM (NVIDIA RTX3090) attached to a 16-core/32-thread CPU (AMD Ryzen 7950X) with 64GB RAM. All but the largest models can be trained in under a day on a machine of this specification. We provide emissions estimates for trained models in the main paper.

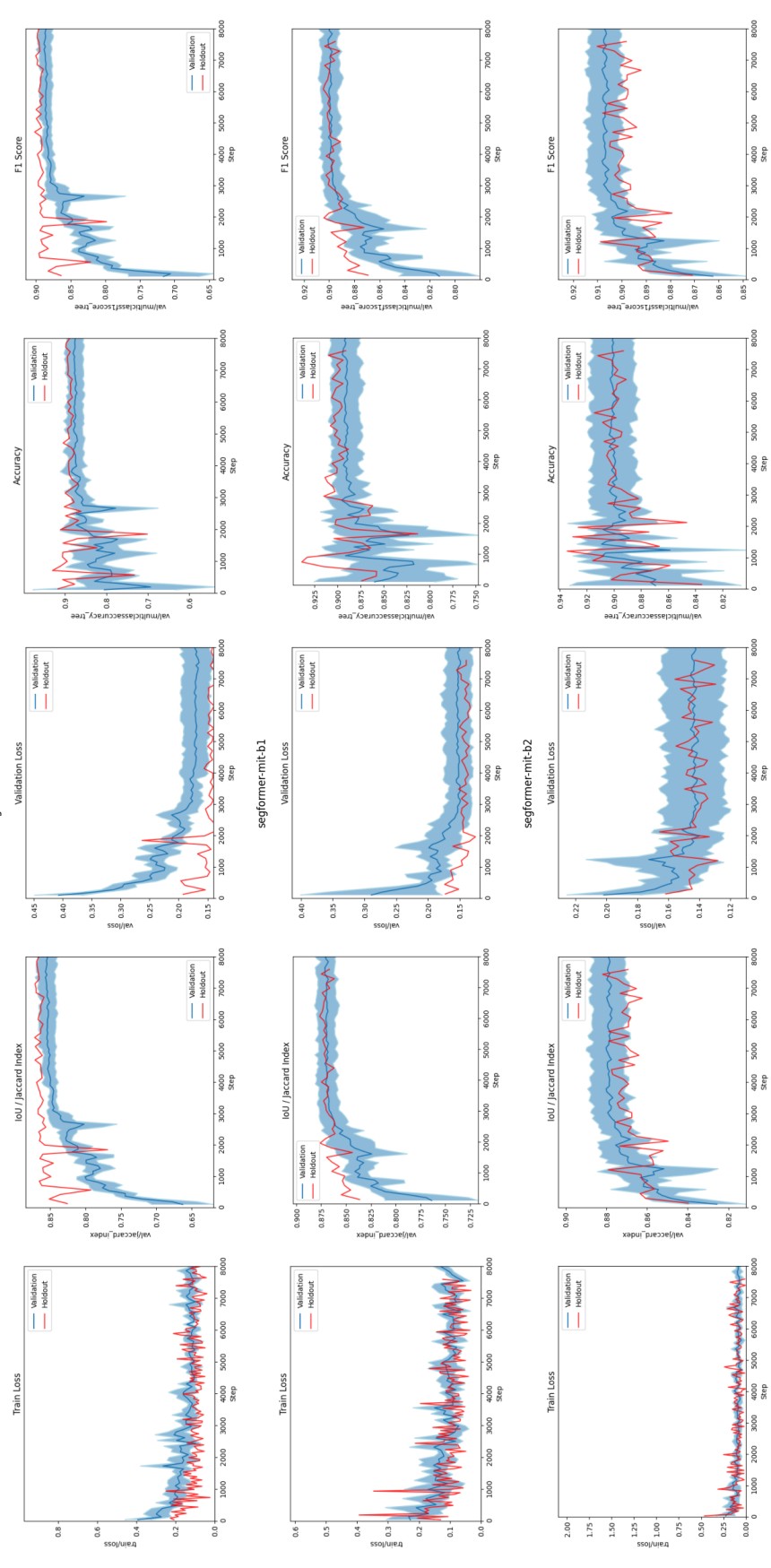

Figure 3: Training/validation curves for SegFormer mit-b0 to mit-b2

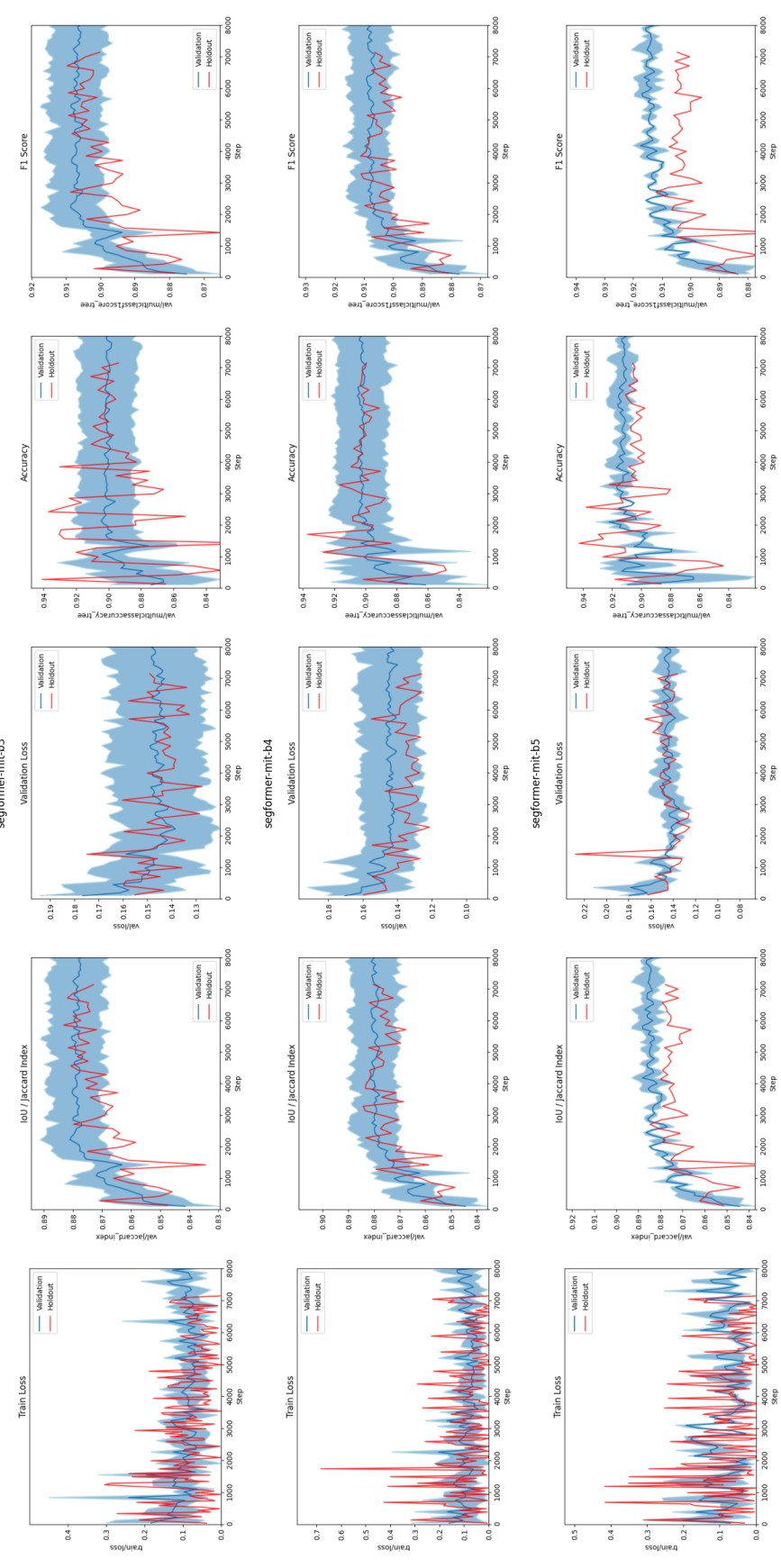

Figure 4: Training/validation curves for SegFormer mit-b3 to mit-b5

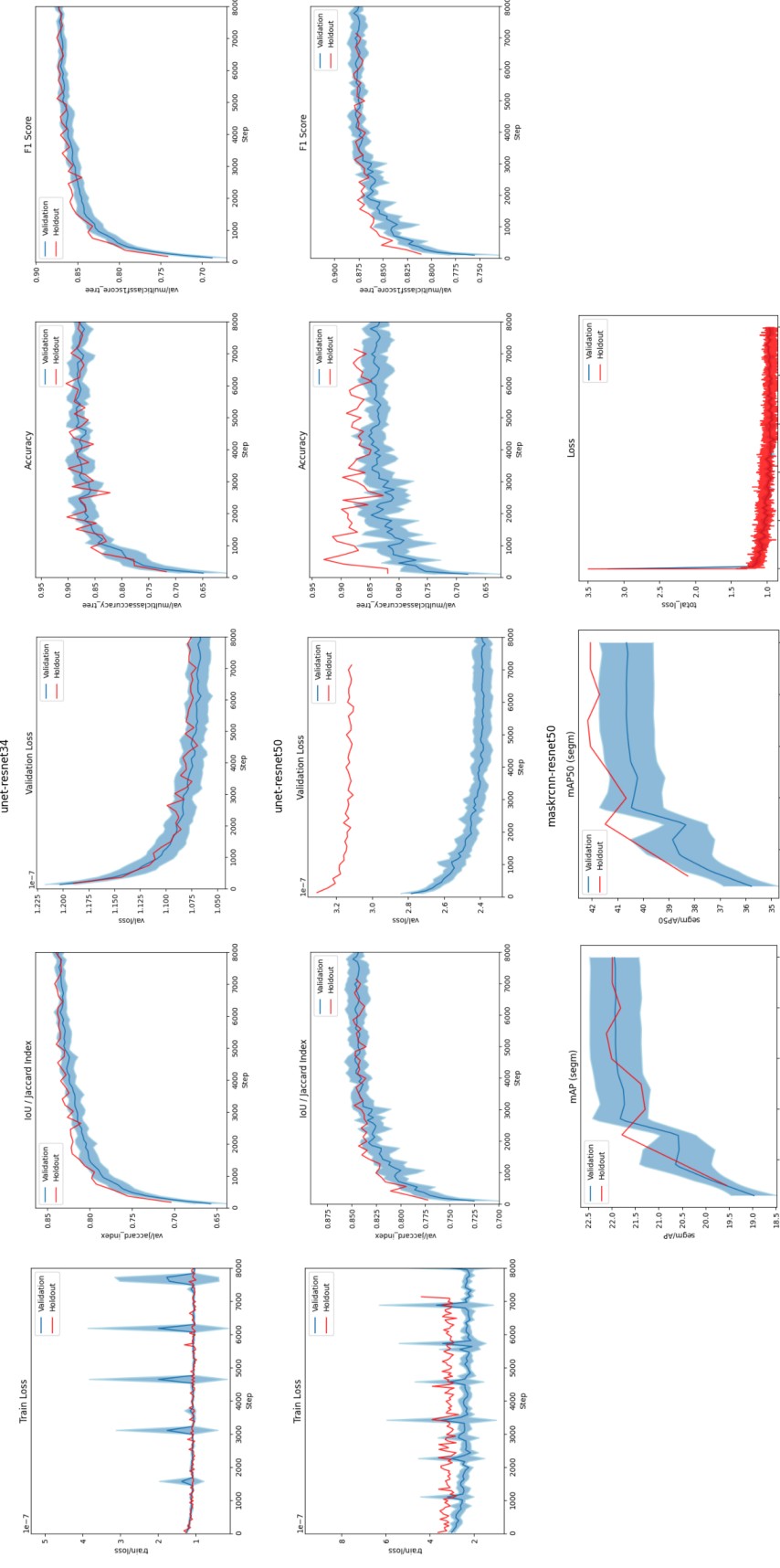

Figure 5: Training/validation curves for UNet models and Mask-RCNN

| Model Variant | Train time (hrs) | Energy Usage (kWh) | CO2 Estimate (kg) |
|---|---|---|---|
| Mask-RCNN ResNet50 | 126 | 44.1 | 19 |
| Segformer MIT-b0 | 33 | 11.6 | 5 |
| Segformer MIT-b1 | 42 | 14.7 | 6.35 |
| Segformer MIT-b2 | 75 | 26.2 | 11.3 |
| Segformer MIT-b3 | 108 | 37.8 | 16.3 |
| Segformer MIT-b4 | 126 | 44.1 | 19 |
| Segformer MIT-b5 | 150 | 52.5 | 22.5 |
| UNet ResNet34 | 33 | 11.6 | 5 |
| UNet ResNet50 | 90 | 31.5 | 13.6 |
| **Total** | **783** | **274.1** | **118.1** |

Table 4: Emissions estimates for model training. We assume 6 runs per model (i.e. holdout plus 5-fold cross validation) and 350 W system power consumption.

**Evaluation**

Models were evaluated on the cross-validation and test splits defined by OAM-TCD. Our pipeline contains evaluation code to compare semantic segmentation output against raster ground truth (e.g. a LIDAR-derived CHM) and instance segmentation output against polygon or keypoint ground truth.

## 3   Pipeline

Our training and prediction pipeline is hosted on GitHub under an Apache 2.0 license: `https://github.com/restor-foundation/tcd`. The pipeline supports prediction on large orthomosaics via image tiling. During prediction, images are split into overlapping tiles (by default 1024 px with an overlap of 256 px). Each tile is passed through the model and the results are cached to disk; the pipeline supports batched inference for both instance and semantic segmentation models. End-user documentation for pipeline usage is stored in the repository.

**Results caching and output**    In order to support predictions over out-of-memory datasets, dataloading and caching during inference are lazy. The pipeline exploits windowed reading and writing supported by GeoTIFFs so that RAM usage is limited to the batch of tiles currently being processed. Results are incrementally stored to a tiled GeoTIFF (for semantic segmentation) or streamed to a shapefile (for instance segmentation). We recommend that virtual rasters (`vrt` files[2]) are used for large inputs.

**Semantic segmentation post processing**    Tile merging follows the approach of (11). Half the overlap region for each tile is discarded and the central region is stored to the output cache; we retain predictions at image edges where there is no overlap. We find that SegFormer models have slightly poorer agreement between tile edges, perhaps due the weaker translational invariance in transformer-based models, compared to the convolutional backbone(s) used in UNet + ResNet.

**Instance segmentation post processing**    Tree instances that intersect the tile boundary are discarded, but we retain group/canopy predictions. Non-max suppression is performed by default on each tile, separately. After inference is complete, a dissolve operation is performed on all instances to identify overlapping predictions. We then heuristically remove tree polygons that contain multiple centroids from other instances, and then make a final merge decision based on a polygon IoU threshold. Instances are stored in an R-Tree for efficient processing.

**Region-based filtering**    Results can be optionally filtered via an input geometry, such as a shapefile. This is necessary for reporting results that are constrained to a region of interest, for example measuring canopy cover for a particular plot of land.

---

[2]VRT - GDAL Virtual Format, `https://gdal.org/drivers/raster/vrt.html`, accessed 1 Jun 2024

**Libraries used**    Our pipeline relies on several Python packages for geospatial processing which we acknowledge here: `rasterio` (8) for general image loading and manipulation (which heavily depends on GDAL (4)); `rtree` (7) for spatially efficient indexing and intersection calculations; shapely for polygon and other geometry processing (5; 9); `fiona` (6) for input geometry handling (i.e. Shapefiles, GeoJSON support).

## 3.1   Sample Model Predictions

Figure 6 and Figure 7 show complete overviews of model predictions shown in the main paper, using our pipeline.

We processed the city of Zurich using our `segformer-mit-b5` model as a demonstration. Switzerland (Swisstopo) releases imagery over the entire country, but only 1/3 of the cantons are captured each year and the timing of surveys is not consistent (so tree cover is not directly comparable between subsequent captures of the same site). Using a single NVIDIA RTX3090 for compute, the prediction took approximately 1.5 hours for approximately 15k ha of imagery (the image extent is 19.6k ha, but we automatically skip inference on empty tiles).

## 4   Further Suggestions for Usage

**Fine-tuning**    Since we distribute weights that are compatible with widely used training frameworks, we expect they can be effectively fine-tuned on other tree detection/classification tasks. This approach has been demonstrated with DeepForest (16), for example.

**High-resolution panoptic segmentation**    Although OAM-TCD only contains labels for tree cover, it would likely be beneficial to augment the dataset with semantic annotations for other features including buildings, roads and geographic features like water cover.

**Evaluation of foundational models**    There is increasing interest in training foundational models that learn robust representations for geospatial data, given huge repositories of open imagery available. These models can then be used directly (zero-shot) or fine-tuned on downstream tasks. Accordingly, multi-task benchmarks have been proposed (e.g. GEO-Bench (12)); OAM-TCD is potentially a useful addition to this landscape due to its flexible label resolution and excellent geographic diversity

**Surface map filtering**    LIDAR-derived data products are often considered to be a gold standard for forest remote sensing, but classifying point clouds remains a challenging research problem analogous to 2D semantic segmentation. This includes ground extraction, to construct Digital Terrain Models, and classifying points as building, vegetation, etc. In natural environments, once ground has been subtracted, a common assumption is that anything that remains is vegetation. Scans of urban environments must be filtered to discriminate artificial objects as well. High resolution tree canopy maps can be used in conjunction with LIDAR scans to better filter tree-cover compared to point-cloud classification alone.

**Unsupervised clustering**    Since the dataset contains a large number of tree polygons, it is likely that clustering would be effective to isolate tree crowns with distinct morphology, like palms. As instance segmentation models return masks instead of bounding boxes, predictions are usually less contaminated with background pixels and the mask itself contains geometric information about the detected object. This also makes the outputs more amenable for training species classifiers on prediction crops, if species are known. We provide an example tool for clustering using BioCLIP embeddings in our pipeline.

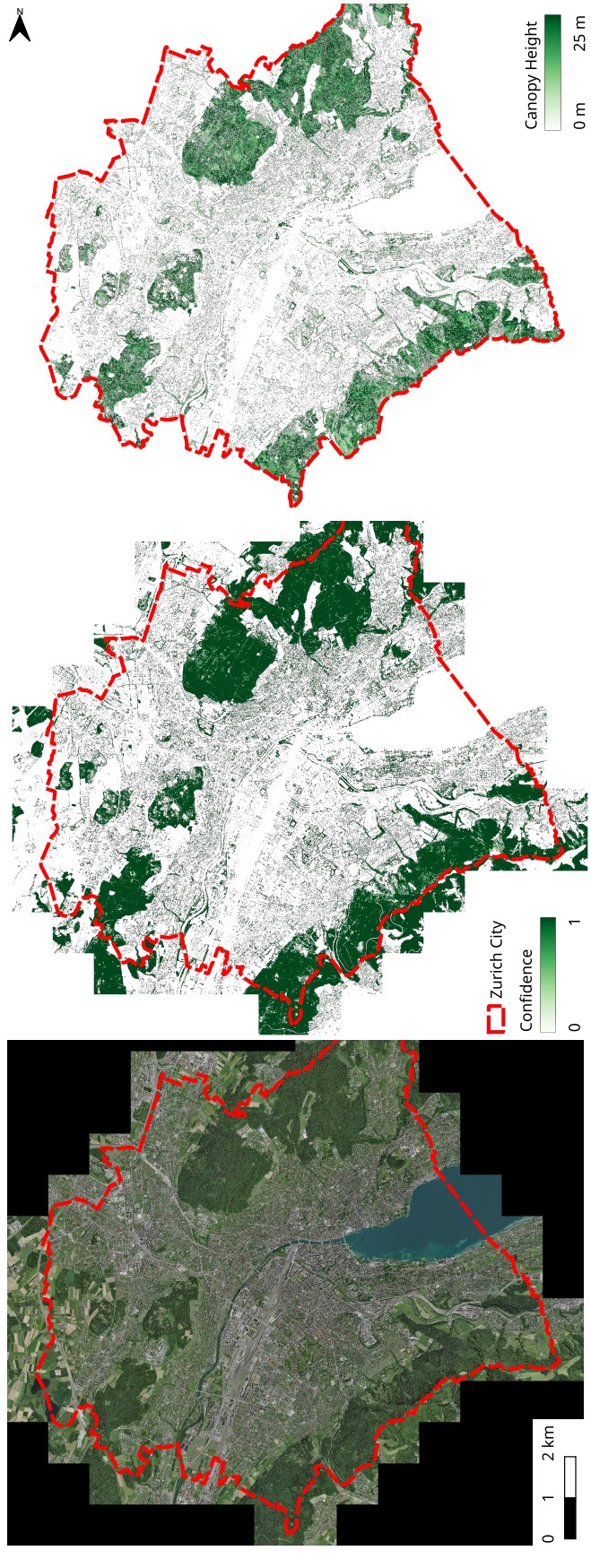

Figure 6: Semantic segmentation predictions for the city of Zurich, with the municipal city boundary marked. Left: 10 cm RGB orthomosaic provided by the Swiss Federal Office of Topography swisstopo/SWISSIMAGE 10 cm (2022), Middle: model predictions, Right: LIDAR CHM (Gruen Stadt Zurich)

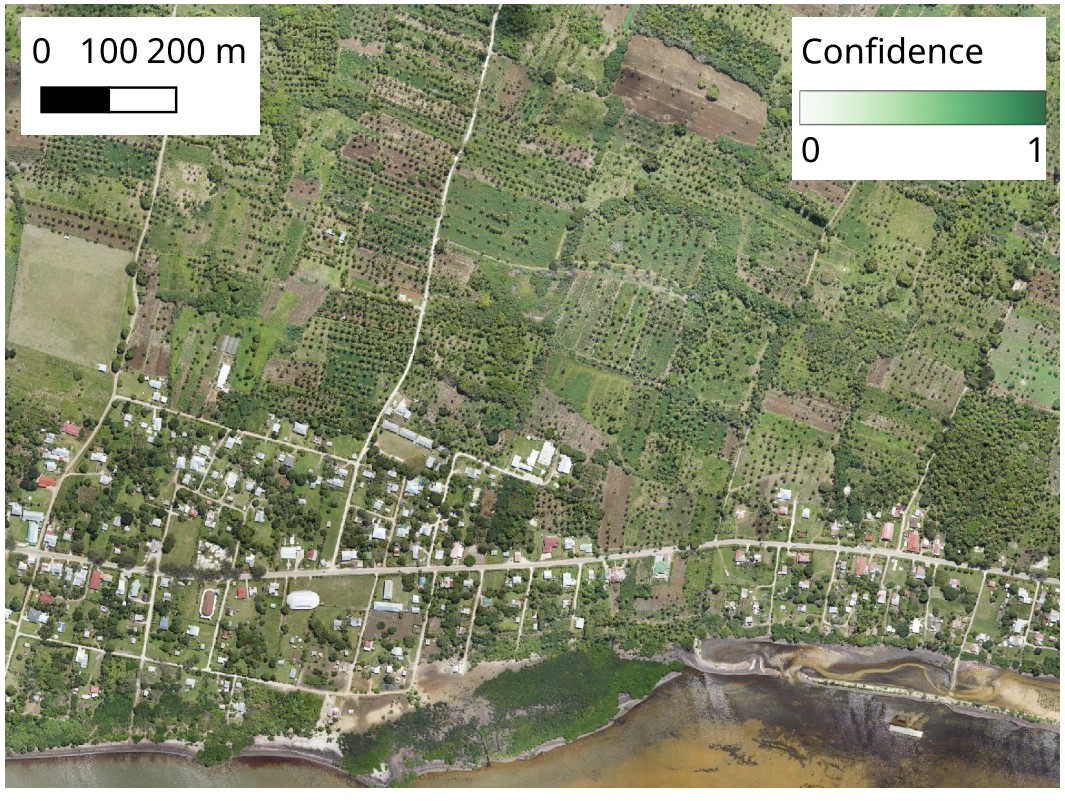

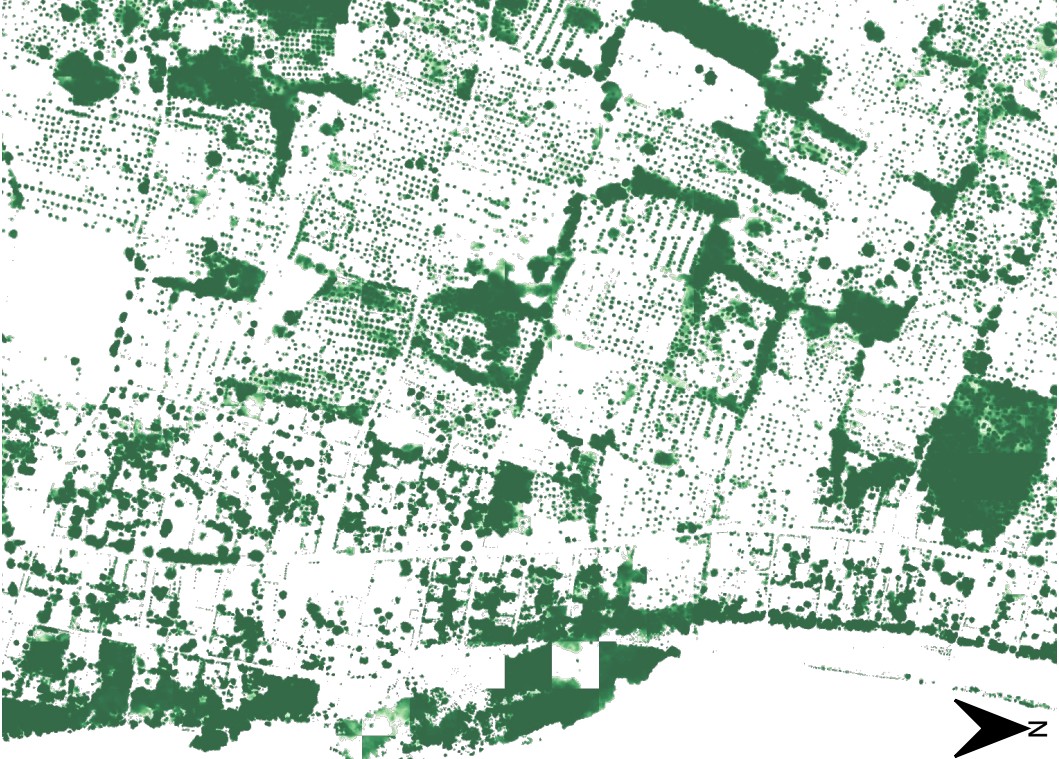

Figure 7: Semantic segmentation predictions for the WeRobotics Open AI challenge image over the Kingdom of Tonga, using the `restor/tcd-segformer-mit-b5` model. Individual palm trees are clearly segmented. Some uncertain predictions are visible in the lower region of the image near the coast - identifiable as missing/inconsistent tiles.