# OpenReview forum: "OAM-TCD: A globally diverse dataset of high-resolution tree cover maps"
_NeurIPS.cc/2024/Datasets_and_Benchmarks_Track — NeurIPS 2024 Track Datasets and Benchmarks Poster_

### Official Review · Reviewer_rHN3 · 2024-07-24
**A novel globally diverse high-resolution dataset for individual tree crown delination**

**Rating:** 9
**Confidence:** 5

**Review:**

Overall, this dataset significantly improves upon prior related datasets. The authors provide extensive experiments, including assessing out-of-domain performance, and open-source all data, models, code, and a pipeline to operate on large orthomosaic geotiffs. The authors further take on the annotation cost for the community to benefit from this high-quality dataset. This work is relevant to the D&B track and I highly recommend this paper for acceptance.

**Strengths:**

- The authors develop a novel dataset which address deficiencies in other related datasets
- The authors provide baselines for canonical CNN and MaskRCNN models as well as SegFormer and release these pretrained models.
- The experiments and analysis are significant and thorough
- The authors take on the annotation planning and cost for the community to benefit from.
- All models are trained on a consumer GPU RTX 3090 which allows for easier reproducibility and/or fine-tuning by the community.
- The dataset and labels are made available in an ML-ready format
- The authors assess global and geographic diversity which is important for remote sensing.

**Additional Feedback:**

N/A

**Clarity:**

The paper is well written and organized. Each section is well defined and does not contain filler.

**Correctness:**

The claims in the dataset appear to be valid and correct. The experiments are well thought out and follow standard practice for semantic and instance segmentation. The authors further take care to evaluate on out-of domain data to assess generalization performance which is much needed when dealing with remote sensing datasets and modeling. The authors further perform experiments using k-fold cross validation and report mean/std across runs which further solidifies their results.

**Documentation:**

The dataset is well documented, is available on HuggingFace in a easily usable format and sufficient detail is provided in the main paper, supplementary material, and the official dataset website [1]

[1] https://restor-foundation.github.io/tcd/dataset/

**Ethics:**

No ethical concerns.

**Limitations:**

The authors sufficiently address limitations of this work and societal impact in the corresponding section.

**Opportunities For Improvement:**

One recommendation would be to assess how the model performs at different resolutions. Does this model potentially transfer to lower resolutions like 30-50cm if images and masks are downsampled as an augmentation during training. This would be useful as 10cm/px aerial imagery is not always available but 30-50cm aerial and satellite imagery is in abundance globally.

**Relation To Prior Work:**

Other related tree delination/detection/segmentation datasets generally suffer from being low-resolution, weakly-labeled or noisy, not globally diverse, or using imagery with significant warping and motion blur issues like that of the NEON imagery [1]. The OAM-TCD dataset clearly addresses these issues and presents a high-quality novel dataset for the task of tree crown delination.

[1] https://www.neonscience.org/

**Summary And Contributions:**

The authors propose a new dataset, OAM-TCD which contains high-resolution imagery from OpenAerialMap and manually annotated tree pixelwise annotations.

Existing tree crown delination datasets are typically either too low-resolution, contain noisy labels, contain low-quality imagery, or contain only bounding box labels as opposed to pixelwise annotations. The authors perform thorough baselines on the proposed dataset and other benchmark datasets to assess performances both in-domain and out-of-domain. The dataset, code, and pipeline for processing large orthomosaics is made available and easy to use in ML ready formats. This dataset is has the potential to be very valuable for the ecological remote sensing community.

---

> ### Author Rebuttal · Authors · 2024-08-14
>
> Thank you for providing a comprehensive and very positive review. We tried to ensure that our work is well documented and are glad that you found it to be.
>
> **Regarding training on different resolutions.**
>
> To support this work (training) in the future we are modifying our dataset generation script to include a target GSD parameter so it can be used to export arbitrarily scaled annotation masks. As both our out-of-domain tests use aerial imaging vs a mixture of sensors in the dataset, there is some evidence that our models generalise well to different sensors. Transferring to satellite imagery will likely be complex, as there are considerations around point spread functions, spectral bands, atmospheric correction/haze, clouds, minimum object size after scaling etc. Due to these additional questions and the analysis required, we felt it would be more appropriate to leave this for future work (but to at least make it straightforward to generate datasets at different resolutions).
>
> Our guess is that, zero-shot without re-training, the existing models may work reasonably well at 20-30 cm. Since the models have no explicit knowledge of scale there will certainly be some false predictions and this will get worse as resolution decreases. The semantic segmentation models will probably fare better than instance detectors.
>
> As an aside, one avenue for future research would be to attempt to train models that are explicitly scale-aware. We have not explored approaches to include this sort of prior knowledge, but it could presumably include things like pixel scale, biome index, height from photogrammetry if available etc.
>
> **Update 19/8/24** - we have trained some segformer-mit-b0 models on 50 cm rescaled imagery following the cross validation approach in the paper (training is quite fast due to the smaller image size). We will upload some preliminary results this week. Typical F1 scores (tree class) are around 0.8 which is probably due to naive down-sampling of the data. We did not perform any post-scaling filtering e.g to remove hot pixels in the masks that correspond to very small annotations and this may have contributed to the error.

---

### Official Review · Reviewer_heBU · 2024-07-24

**Rating:** 5
**Confidence:** 3
**Correctness:** Strengths & Weaknesses
**Clarity:** Yes

**Review:**

Strengths & Weaknesses

**Strengths:**

1. OAM-TCD addresses the lack of open, globally diverse, high-resolution tree detection datasets. It captures a wide range of tree species, morphologies, and environments by sourcing images from various locations through OpenAerialMap. This makes it a valuable resource for developing and evaluating tree detection models.

2. The authors rigorously test the utility of OAM-TCD by evaluating baseline models on both the dataset itself and completely independent data from Zurich and the WeRobotics Open AI Challenge. This thorough testing approach demonstrates the robustness and generalization capabilities of the models.

3. The work exempliﬁes strong open science practices by making the dataset, trained models, and code publicly available under permissive licenses. This transparency enables reproducibility, encourages further development, and contributes to the broader advancement of tree detection methods and their applications.

**Additional Feedback:**

Strengths & Weaknesses

**Documentation:**

Yes

**Limitations:**

Yes

**Opportunities For Improvement:**

1. While the OAM-TCD dataset is geographically diverse, there may still be some biases or underrepresentation of certain regions or biomes. The authors acknowledge that some areas, such as Northern Africa and Northeast Asia, have limited coverage in the dataset. This could potentially impact the performance of models trained on OAM-TCD when applied to these underrepresented regions, as the models may not have learned features speciﬁc to the trees and environments in those areas.

2. The paper could beneﬁt from a more detailed discussion of the challenges and limitations encountered during the annotation process. The authors mention that annotators were instructed to label individual trees where possible and mark groups of trees as "closed canopy" when individual delineation was not feasible. However, there may be some inconsistencies or subjectivity in these decisions across diﬀerent annotators. Providing more insights into the quality control measures, inter-annotator agreement, and potential impacts of these annotation choices on the trained models would strengthen the paper and help users better understand the dataset's limitations.

**Relation To Prior Work:**

Yes

**Summary And Contributions:**

In their paper, Veitch-Michaelis et al. introduce OAM-TCD, a novel open-access dataset designed to advance the state-of-the-art in individual tree crown delineation from high-resolution aerial imagery. The dataset comprises 5072 images at 10 cm/px resolution, sourced from OpenAerialMap, with instance masks for over 280,000 individual trees and 56,000 tree groups. By sampling imagery from diverse geographic locations, OAM-TCD captures a wide range of tree morphologies and environments, addressing a key limitation of existing datasets. The authors demonstrate the utility of OAM-TCD by training and evaluating baseline models for both semantic and instance segmentation tasks. Cross-validation results and comparisons with existing datasets show strong performance. Notably, the models also perform well on independent aerial imagery from Zurich and the WeRobotics Open AI Challenge, showcasing their potential for real-world applications.

---

> ### Author Rebuttal · Authors · 2024-08-14
>
> Thank you for your review, we appreciate that you commended the real-world utility of the dataset, rigorous experimentation and our commitment to open research practice.
>
> Given the relative strengths and weaknesses that you’ve highlighted, we were surprised to see a below-acceptance rating (5). Would you be willing to provide further clarification on this decision, for example additional information or experiments that would be beneficial?
>
> Regarding the opportunities for improvement:
>
> 1. **Dataset diversity** From the locations mentioned in your comment, we assume you are referring to the coverage map in Figure 2 (where these regions are given as examples). In the supplementary information, we discuss other measures of diversity that can be used to evaluate a remote sensing dataset of this kind. We would direct the reviewer to the supplementary information where this is described in detail - specifically we provide breakdown of different biomes in the dataset and a discussion of potential biases and limitations.
>
>    Specifically, we use geographic distribution and biome coverage. These are not exhaustive, but are simple and interpretable proxies for dataset diversity. In both cases, OAM-TCD is shown to be one of the most diverse open datasets of its kind. There are limitations to both measures \- some locations are more or less biodiverse than others, and biomes are defined by a geospatial boundary (there is no guarantee that an image represents a natural ecosystem).
>
>    Comparing our biome distribution against the locations of tracked restoration projects in [1] (see Figure 2, against the table in the supplementary info), there is a good correlation including in sites that are under-represented; this suggests our dataset is actually well positioned to support (at least) conservation and restoration site monitoring.
>
>    The fact that our models perform well on out-of-domain data is supportive of the fact that our models generalise. We acknowledge that there are deficiencies in coverage that we will try to improve in future dataset releases; this will also be guided by users testing models on their own imagery. Due to the high annotation cost, it would make sense to  prioritise areas that are (a) lacking in dataset coverage, and (b) would be beneficial for the community. For example the dataset has relatively poor boreal and mangrove coverage, but here we would likely prioritise acquiring mangrove imagery as those sites tend to be more threatened (and boreal forest mapping is also better represented in the literature by e.g. researchers from Scandinavia with access to high quality national aerial imaging).
>
> 2. **Annotation limitations and process**
>
>    We have not performed a statistical analysis of inter-annotator agreement because, while images were reviewed by independent experts, each image was annotated by a single person in most cases. Imagery was reviewed by the authors and other colleagues who have expertise in region-specific forest ecology in cases where there were questions (e.g. is an object a tree or a shrub?). This was a trade-off between obtaining a sufficient quantity of labelled data, the quality of the labels, and the time/financial investment involved. Aerial imagery at this resolution is complex and the annotation cost per image is very high.
>
>    As with the diversity limitations, we feel that the results of our models on independent data support this decision and still allow us to improve labels over time. There is inherently less ambiguity when marking a pixel as covered by a tree versus whether it belongs to a specific individual (or a group). Our semantic segmentation masks are (in our opinion) consistent and our focus for label improvement is at the instance level.
>
>    This is an area that we are continuing to work on, for example our citizen science campaign on Zooniverse aims to provide a confidence indicator for which labels are correct and which are not. We do have a small pool of data that was revised by a second independent annotator pool, but at this time the analysis is not mature and we plan to release these additional labels at a later date.
>
> [1]  Crowther et al. _Restor: Transparency and connectivity for the global environmental movement_, One Earth, Volume 5, Issue 5, 20 May 2022, Pages 476-481

---

### Official Review · Reviewer_AXXa · 2024-07-27
**280k Trees Manually Segmented from Crowd-Sourced High-Resolution Aerial Imagery Across the Globe**

**Rating:** 7
**Confidence:** 5
**Clarity:** Yes.

**Review:**

The presented manuscript is written in plain English. It contains relevant figures and tables to support the claims of the work. The main text is densely covered by relevant references to existing literature. Results are presented clear and honest.

The dataset proposed, OAM-TCD, derives from globally sampling of the OpenAerialMap (OAM) database. Approximately $25k was invested in having human annotators outline the extent of 280k trees and 56k groups of canopy based on OAM data. For testing semantic segmentation models, the authors utilize high-quality tree data from Zurich supported by a LiDAR survey. In addition, testing of the models on a tree localization data challenge hosted by OAM is carried out.

The presented paper is a solid contribution. Since the annotation of OAM-TCD is human-labeled compared to naive LiDAR thresholding, the dataset is a valuable piece for the "Earth Observation with AI" community to develop robust tree segmentation models. In fact, due to the fluctuation of sensor quality in a plurality of image sensors, OAM-TCD may well serve as input to geospatial foundation models for vegetation.

Two weak spots of OAM-TCD concern:
- 10cm spatial pixel resolution cuts below the most coarse-grained resolution of 1 meter for military applications. Correspondingly, legal (defense) and ethical (surveillance) consequences could have been discussed in more detail.
- The authors trust OAM when it comes to licensing. Also, I wonder: Does the OAM crowdsourcing initiative have consensus from users uploading the overhead imagery to redistribute their work?

**Strengths:**

The manuscript appears extremely clean and to the point.

**Additional Feedback:**

According to https://neurips.cc/Conferences/2024/ReviewerGuidelines the authors are responsible for double-blind reviewing. The manuscript submitted reveals author identities (names and affiliations).

**Correctness:**

To the best of my knowledge, the OAM dataset is sound and all statements in the paper appear honest.

**Documentation:**

- code: https://github.com/Restor-Foundation/tcd (Apache 2.0 license), very well documented
- data: https://huggingface.co/restor/tcd (CC-BY 4.0 license, attention: not yet available?)

ATTENTION: I expect the dataset becomes available upon publication of the work. Without, I would need to downgrade my rating of the work to "5: Marginally below acceptance threshold". @authors: Pls confirm that the manuscript has a typo, i.e. does not share the correct link. I see this publicly available: https://huggingface.co/datasets/restor/tcd

**Ethics:**

The only red flag to me is the military-grade (10cm) geospatial resolution of OAM.

**Limitations:**

Given the nature of OAM, the sampled data is not globally distributed to cover Earth's entire vegetation, cf. Fig2. Tha authors honestly not this fact.

**Opportunities For Improvement:**

minors:
- environmental impact: pls estimate total CO2 consumed for paper
- "data available for third party testing available so" - remove "available"
- "heirarchical" to "hierarchical"
- The authors state: "mapping at the sub-meter scale. This is particularly relevant for assessing sparsely distributed trees outside forest areas which are not visible in low resolution images.". From my experience NAIP overhead imagery allows to resolve single trees on the meter scale for at least urban scenarioes. Perhaps the statement can be reformulated to appear less strong.
-  "the globe at 10-30m resolution; this analogous to NASA’s Landsat program.", two thoughts:
    * "this" to "thus"?, and
    * the multi-spectral Sentinel-2 sensor has pixel sizes varying from 10m to 60m.

**Relation To Prior Work:**

yes, well documented

**Summary And Contributions:**

The work presents a carefully collected open-source tree segmentation dataset based on 10cm-resolution overhead imagery. The aerial imagery has been sampled from the OpenAerialMap project covering about 20k ha of land. UNets and SegFormers serve as benckmark baseline for semantic segmentation of trees. The trained models are tested on detection of urban forests in Zurich where high quality verification of the results on the ground is possible.

---

> ### Author Rebuttal · Authors · 2024-08-14
>
> **Documentation**
>
> Yes, you are correct that there is a typo in the dataset URL; it is already publicly available. This looks like a mix-up between our HuggingFace [organisation page](http://huggingface.co/restor) (links to all models and the dataset) and the direct [link](http://huggingface.co/datasets/restor/tcd) to the dataset itself. Our apologies\!
>
> **Total CO2 Emissions Estimates**
>
> Below is a summary table of the power consumption and emission info required to reproduce model results, which we will include in the supplementary information; we’ll update the main paper to include the total estimate. Individual emissions figures will be added to the model datasheets on HuggingFace.
>
> We assume the default carbon efficiency value (0.432 kg/kWh) given by Lacoste et al.  **GPU**: NVIDIA RTX 3090, power target 200 W. **System**: AMD Ryzen 7950X (TDP 170 W). CPU \+ peripherals estimated usage at 150W during training.
>
> | Model Variant | Power consumption | Training Hours (single run) | Number of models trained | Total Energy Usage (kWh) | CO2 Estimate (kg) |
> | :---- | :---- | ----- | ----- | ----- | ----- |
> | **Mask-RCNN ResNet50** | | 21 | 6 (holdout plus 5-fold cross validation) | 44.1 | 19 |
> | **Segformer MIT-b0** |  | 5.5 |  | 11.6 | 5 |
> | **Segformer MIT-b1** |  | 7 |  | 14.7 | 6.35 |
> | **Segformer MIT-b2** |  | 12.5 |  | 26.2 | 11.3 |
> | **Segformer MIT-b3** |  | 18 |  | 37.8 | 16.3 |
> | **Segformer MIT-b4** |  | 21 |  | 44.1 | 19 |
> | **Segformer MIT-b5** |  | 25 |  | 52.5 | 22.5 |
> | **UNet ResNet34** |  | 5.5 |  | 11.6 | 5 |
> | **UNet ResNet50** |  | 15 |  | 31.5 | 13.6 |
> | **Total** | **350 W** | **130.5** | **54** | **783** | **338** |
>
> Conservatively we estimate double the total training time for experimentation, but the majority (\~80%) of the compute was devoted to cross-validation, performed once the training pipeline was stable. The time/cost for training a single model, for example a user fine-tuning on new data, is much smaller.
>
> **General comments**
>
> r.e. NAIP; we agree and will make the statement less strong. For object/instance detection, the additional resolution is quite beneficial as trees may only be a few pixels across in lower resolution data. For semantic segmentation it’s less of a factor, as evidenced by literature that uses Planet (3 m).
>
> Platform/sensor information is in principle available in the metadata for each image, but it relies on users providing detailed information when they submit imagery.
>
> **Ethics surrounding image resolution**
>
> We acknowledge that there are ways in which one could use our models for unethical purposes and will add details in the manuscript (or supplementary material, space depending). In most cases, unethical use of our work requires intent. One question is: what advantage do our models give to malicious or unethical users over possession of very high resolution imagery alone? A tangential result of our work is that we provide a concrete data point for cost:performance which may encourage users to replicate our work (or expand upon it) for unethical applications.
>
> In terms of availability of high resolution imagery: numerous governmental organisations now provide sub-meter and even \< 50 cm imagery with little-to-no access limitations; in the case of several European countries, 10-25 cm scale imagery is available (Switzerland, France, Estonia, Lichtenstein, etc.). Obtaining *current/up-to-date* imagery of a specific location at this resolution generally requires financial investment, but is basically accessible to those who can afford it (i.e. satellite tasking) and some applications like illegal land exploitation/logging may not even require 10 cm imagery.
>
> Military and surveillance applications are concerning, given the now widespread use of UAVs/drones in modern theatres of war. On the other hand, it might be financially feasible to simply pay staff to inspect and label/review imagery; access to on-demand surveillance imagery (such as reconnaissance satellites or similar assets like high resolution SAR) is likely a more important factor. While it’s true that our models may facilitate better mapping for very high resolution imaging (albeit for a single class), drone image analysis is also already offered as a service by several companies.
>
> For fairness, we should also consider the benefits that are gained by using high resolution data. For some conservation applications like detecting planting holes or seed/saplings; species identification[1]  based on leaf morphology, etc; even 10 cm/px is potentially too coarse. Drone/UAV imaging is one of the most accessible and affordable ways that stakeholders can monitor land, particularly in regions where satellite/aerial imagery coverage is poor. Our aim is to facilitate tree mapping at a higher resolution than is currently possible, ultimately to improve transparency in the sector.
>
> **OAM Licensing**
>
> In addition to OAM’s mission statement/”about” pages, there is fairly clear copy on the homepage that states:
>
> \> *OpenAerialMap is an open service to provide access to a commons of openly licensed imagery and map layer services.*
>
> And in the upload form:
>
> *\> By submitting imagery to OpenAerialMap, you agree to place your imagery into the Open Imagery Network (OIN). Except when permitted by the OpenStreetMap exception (see below), all imagery contained in OIN is licensed CC-BY 4.0, with attribution as "© OIN contributors", and specific additional SA/NC conditions if selected upon upload.*
>
> With this in mind, we trust that image uploaders understand that their imagery is (or will be) permissively licensed via the OIN. Uploaders do have the option to include \-NC or \-SA restrictions although this is not clearly stated on the OAM homepage. We distribute these images in separate repositories to avoid concerns about unintentionally mixing CC licenses.
>
> [1]  Species identification, e.g. locating valuable trees to log, is not currently possible using OAM-TCD.

---

### Author Rebuttal · Authors · 2024-08-14

Dear Reviewers,

Thank you for taking the time to write comprehensive reviews of our work. We are encouraged by the positive feedback and will integrate your proposed changes. Since submitting the initial draft, we have taken the opportunity to polish various sections of the manuscript with improved figures and minor revisions to the text for readability and clarity.

Please check the rebuttal section of your reviews for detailed responses to your individual suggestions for improvement.

Summary of changes:

* Fixed typos/grammatical mistakes
* The sentence on Sentinel 2 is reworded to better reflect the capabilities and revisit time of the platform.
* Updated prediction images to be generated via script for repeatability; the underlying predictions are unchanged, but the viewpoint in the figures may be a little off the originals.
* Included full-page predictions for Zurich and Tonga in the supplementary information.
* Updated section on ethics/potential for negative use cases (see AXXA rebuttal for high level discussion).
* Full CO2 emission estimates broken out by model variant (see AXXa rebuttal).
* Usability improvements to the public codebase, particularly command line scripts for common tasks.
* Modified dataset generation script to support downscaling/exporting annotation masks to a target Ground Sample Distance (GSD).
* Corrected a couple of URLs (dataset direct link) and also to our Zooniverse project as mentioned in the conclusion (the link provided was an older version that was not activated)
* Removed the reviewer overview page in the supplementary info. API keys are no longer required to obtain the dataset/models.
* Per reviewer rHN3's suggestion, we have trained some preliminary semantic segmentation models (segformer-mit-b0) on downscaled imagery with F1 (tree) scores around 0.8; we will update the documentation with information about dataset scaling and training.
* Added a reference to the [FLAIR 2](https://github.com/IGNF/FLAIR-2) dataset (NeurIPS '23) which offers similar resolution (20 cm) and annotation completeness, but only over France and per-pixel labels only. In the future it should be straightforward to release co-registered imagery from e.g. Sentinel 2 which may aid prediction.

Kind regards,
OAM-TCD Authors

NB: per the Datasets and Benchmarks track guidelines, single-blind submission is acceptable, which accommodates existing public release of the dataset/models that would be difficult to anonymise. This does differ from the submission rules for the main conference track.

---

### Decision · Program_Chairs · 2024-09-26

**Decision:**

Accept (Poster)

**Comment:**

Three reviewers provided comments, but unfortunately no-one reacted to the very comprehensive rebuttals. This area chair therefore weighed in their own analysis of paper and rebuttal. Two reviewers are very positive and the comments by reviewerAXXa are well responded to by the authors. The only below acceptance reviewer has very minor points in their review, and their comment does not seem to provide arguments sufficient to reject the article. For this reasons I decided to accept the paper. Congratulations!